# DATEFORMER: TRANSFORMER EXTENDS LOOK-BACK HORIZON TO PREDICT LONGER-TERM TIME SERIES

## ABSTRACT

Transformers have demonstrated impressive strength in long-term series forecasting. Existing prediction research mostly focused on mapping past short sub-series (lookback window) to future series (forecast window). The longer training dataset time series will be discarded, once training is completed. Models can merely rely on lookback window information for inference, which impedes models from analyzing time series from a global perspective. And these windows used by Transformers are quite narrow because they must model each time-step therein. Under this point-wise processing style, broadening windows will rapidly exhaust their model capacity. This, for fine-grained time series, leads to a bottleneck in information input and prediction output, which is mortal to long-term series forecasting. To overcome the barrier, we propose a brand-new methodology to utilize Transformer for time series forecasting. Specifically, we split time series into patches by day and reform point-wise to patch-wise processing, which considerably enhances the information input and output of Transformers. To further help models leverage the whole training set's global information during inference, we distill the information, store it in time representations, and replace series with time representations as the main modeling entities. Our designed time-modeling Transformer—Dateformer yields state-of-the-art accuracy on 7 real-world datasets with a 33.6% relative improvement and extends the maximum forecast range to half-year.[1]

## 1 INTRODUCTION

Time series forecasting is a critical demand across many application domains, such as energy consumption, economics planning, traffic and weather prediction. This task can be roughly summed up as predicting future time series by observing their past values. In this paper, we study *long-term forecasting* that involves a longer-range forecast horizon than regular time series forecasting.

Logically, historical observations are always available. But most models (including various Transformers) infer the future by analyzing the part of past sub-series closest to the present. Longer historical series is merely used to train model. For short-term forecasting that more concerns series local (or call short-term) pattern, the closest sub-series carried information is enough. But not for long-term forecasting that requires models to grasp time series' global pattern: overall trend, long-term seasonality, etc. Methods that only observe the recent sub-series can't accurately distinguish the 2 patterns and hence produce sub-optimal predictions (see Figure 1a, models observe an obvious upward trend in the zoom-in window. But zoom out, we know that's a yearly seasonality. And we can see a slightly overall upward trend between the 2 years power load series). However, it's impracticable to thoughtlessly input entire training set series as lookback window. Not only is no model yet can tackle such a lengthy series, but learning dependence from therein is also tough. Thus, we ask: *how to enable models to inexpensively use the global information in training set during inference?*

In addition, the throughput of Transformers (Zhou et al., 2022; Liu et al., 2021; Wu et al., 2021; Zhou et al., 2021; Kitaev et al., 2020; Vaswani et al., 2017), which show the best performance in long-term forecasting, is relatively limited, especially for fine-grained time series (e.g., recorded per 15 min, half-hour, hour). Given a common time series recorded every 15 minutes (96 time-steps per day), with 24GB memory, they mostly fail to predict next month from 3 past months of series, even if

---

[1]Code will be released soon.

they have struggled to reduce self-attention's computational complexity. They still can't afford such a length of series and thus cut down lookback window to trade off a flimsy prediction. If they are requested to predict 3 months, how do respond? These demands are quite frequent and important in many application fields. For fine-grained series, we argue, it has reached the bottleneck to extend the forecast horizon through improving self-attention to be more efficient. So, in addition to modifying self-attention, *how to enhance the time series information input and output of Transformers?*

We study the second question first. Prior works process time series in a point-wise style: each time-step in time series will be modeled individually. For Transformers, each time-step value is mapped to a token and then calculated. This style wears out the models that endeavor to predict fine-grained time series over longer-term, yet has never been questioned. In fact, similar to images (He et al., 2022), many time series are natural signals with temporal information redundancy—e.g., a missing time-step value can be interpolated from neighboring time-steps. The finer time series' granularity, the higher their redundancy and the more accurate interpolations. Therefore, the point-wise style is information-inefficient and wasteful. In order to improve the information density of fine-grained time series, we split them into patches and reform the point-wise to patch-wise

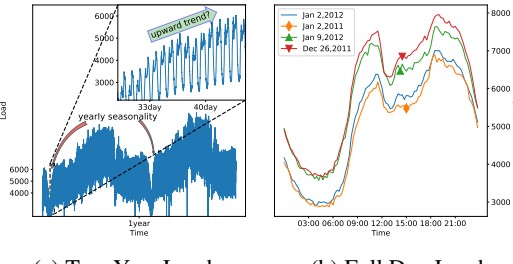

(a) Two Year Load      (b) Full Day Load

Figure 1: (a) depicts two-year power load of an area. (b) illustrates the area's full day load on Jan 2, 2012, a week ago, a week later, and a year ago: compared to a week ago or later, the power load series of Jan 2, 2012, is closer to a year ago, which indicates the day's time semantics is altered to closer to a year ago but further away from a week ago or later.

processing, which considerably reduces tokens and enhances information efficiency. To maintain equivalent token information across time series with different granularity, we fix the patch size as day. We choose the "day" as patch size because it's moderate, close to people's habits, and convenient for modeling. Other patch sizes are also practicable, we discuss the detail in AppendixG.

Nevertheless, splitting time series into patches is not a silver bullet for the first question. Even if do so, the whole training set patches series is still too lengthy. And we just want the global information therein, for this purpose to model the whole series is not a good deal. Time is one of the most important properties of time series and plenty of series characteristics are determined by or affected by it. Can time be used as a container to store time series' global information? For the whole historical series, time is a general feature that's very appropriate to model therein persistent temporal patterns. Driven by this, we try to distill training set's global information into time representations and further substitute time series with these time representations as the main modeling entities. But *how to represent time?* In Section 2, we also provide a superior method to represent time.

In this work, we challenge using merely vanilla Transformers to predict long-term series. We base vanilla Transformers to design a brand-new forecast framework named **Dateformer**, it regards day as time series atomic unit, which remarkable reduces Transformer tokens and hence improves series information input and output, particularly for fine-grained time series. This also benefits Autoregressive prediction: less error accumulation and faster reasoning. Besides, to better tap training set's global information, we distill it, store it in the container of time representations, and take these time representations as main modeling entities for Transformers. **Dateformer** achieves the state-of-the-art performance on 7 benchmark datasets. Our main contributions are summarized as follows:

- We analyze information characteristics of time series and propose splitting time series into patches. This considerably reduces tokens and improves series information input and output thereby enabling vanilla Transformers to tackle long-term series forecasting problems.

- To better tap training set's global information, we use time representations as containers to distill it and take time as main modeling object. Accordingly, we design the first time-modeling time series forecast framework exploiting vanilla Transformers—**Dateformer**.

- As the preliminary work, we also provide a superior time-representing method to support the time-modeling strategy, please see section 2 for details.[2]

---

[2]Related works in Appendix B.

## 2 TIME-REPRESENTING

To facilitate distilling training set's global information, we should establish appropriate time representations as the container to distill and store it. In this paper, we split time series into patches by day, so we mostly study a special case of it—*how to represent a date?* Informer (Zhou et al., 2021) provides a time-encoding that embeds time by stacking several time features into a vector. These time features contain rich time semantics and hence can represent time appropriately. We follow that and collect more date features by a set of simple algorithms[3] to generate our date-embedding.

But this date-embedding is static. In practice, people pay different attention to various date features and this attention will be dynamically adjusted as the date context change. At ordinary times, people are more concerned about what day of week or month is today. When approaching important festivals, the attention may be shifted to other date features. For example, few people care that Jan 2, 2012, is a Monday because the day before it is New Year's Day (see Figure 1b for an example). This semantics shifting is similar to the polysemy in NLP field, we call it date polysemy. It reminds us: a superior date-representation should consider the contextual information of date. However, the date axis is extended infinitely. The contextual boundaries of date are open, which is distinct from words that are usually located in sentences and hence have a comparatively closed context. Intuitively, distant dates exert a weak influence on today, so it makes sense to select nearby dates as context. Inspired by this, to encode dynamic date-representations, we introduce **D**ate **E**ncoder **R**epresentations from **T**ransformers (**DERT**), a convolution-style BERT (Devlin et al., 2019).

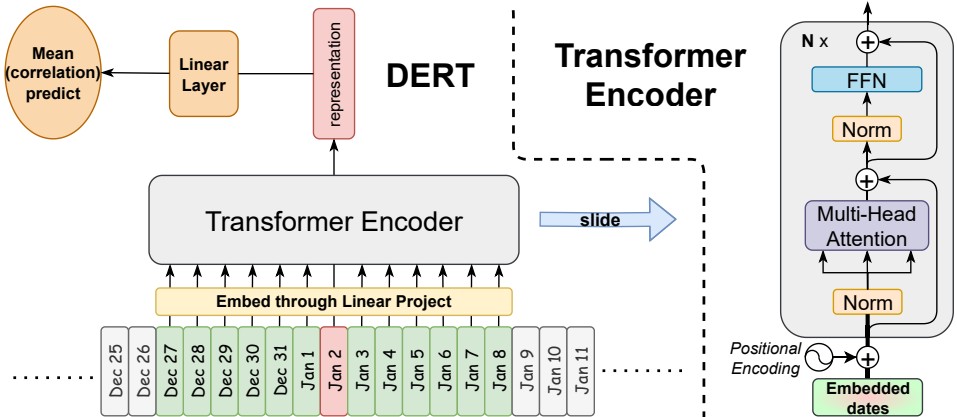

Figure 2: **D**ate **E**ncoder **R**epresentations from **T**ransformers. To encoding the Jan 2, DERT selects date-embeddings of some days before and after it as the contextual padding (the green block).

Concretely, DERT slides a fixed window Transformer encoder on the date axis to select date-embeddings input. The input window contains date-embeddings of: some days before the target day (pre-days padding), the target day to be represented, and some days after the target day (post-days padding). After encoding, tokens of the pre-days and post-days paddings will be discarded, only the target day's token is picked out as the corresponding dynamic date-representation. We design 2 tasks that implemented by linear layers to pre-train DERT.

**Mean value prediction**   Predicting mean values of time series patches is a direct and effective pre-training task, we require DERT to predict the target day's time series mean value by the corresponding date-representation. This task will lose time series patches' intraday information, so we design the latter task to mitigate the problem.

**Auto-correlation prediction**   We observed that many time series display varied intraday trend and periodicity on different kinds of days. Like more steady in holidays, or on the contrary. To leverage it, we utilize the circular auto-correlation in stochastic process theory (Chatfield, 2003; Papoulis & Pillai, 2002) to assess sequences' stability. According to Wiener-Khinchin theorem (Wiener, 1930),

---

[3]see Appendix C for details

series auto-correlation can be calculated by Fast Fourier Transforms. Thus, we Score the stability of the target day's time series patch $\mathcal{X}$ with $G$-length by following equations:

$$
\begin{aligned}
\mathcal{S}_{\mathcal{X}\mathcal{X}}(f) &= \mathcal{F}(\mathcal{X}_t)\mathcal{F}^*(\mathcal{X}_t) = \int_{-\infty}^{\infty}\mathcal{X}_t e^{-i2\pi tf}\,\mathrm{d}t \overline{\int_{-\infty}^{\infty}\mathcal{X}_t e^{-i2\pi tf}\,\mathrm{d}t} \\
\mathcal{R}_{\mathcal{X}\mathcal{X}}(\tau) &= \frac{\mathcal{F}^{-1}(\mathcal{S}_{\mathcal{X}\mathcal{X}}(f))}{\ell_2 Norm} = \frac{\int_{-\infty}^{\infty}\mathcal{S}_{\mathcal{X}\mathcal{X}}(f)e^{i2\pi f\tau}\,\mathrm{d}f}{\sqrt{\mathcal{X}_1^2 + \mathcal{X}_2^2 + \cdots + \mathcal{X}_G^2}} \\
Score &= \mathrm{Score}(\mathcal{X}) = \frac{1}{G}\sum_{\tau=0}^{G-1}\mathcal{R}_{\mathcal{X}\mathcal{X}}(\tau)
\end{aligned}
\tag{1}
$$

where $\mathcal{F}$ denotes Fast Fourier Transforms, $*$ and $\mathcal{F}^{-1}$ denote the conjugate and inverse Transforms respectively. We ask DERT to predict $Score$ of the target day's time series patch by the day's date-representation.

**Pre-training**  We use the whole training set's time series patches to pre-train DERT, each day's time series patch is a training sample and the loss is calculated by:

$$Pretraining\ Loss = \mathrm{MSE}(\mathrm{linearLayer}_1(\boldsymbol{d}), Mean) + \mathrm{MSE}(\mathrm{linearLayer}_2(\boldsymbol{d}), Score)$$

where $\boldsymbol{d}$ denotes the target day's (patch's) dynamic date-representation that produced by DERT, $Mean$ denotes true time series mean value of the day, and $Score$ is calculated by equation 1. Our DERT relies on supervised pre-training tasks, so it's series-specified—different time series datasets need to re-train respective DERT. Note that the 2 tasks are merely employed for pre-training DERT, so they'll be removed after pre-training and not participate in the downstream forecasting task.

## 3  DATEFORMER

We believe that time series fuse 2 patterns of components: global and local pattern. The global pattern is derived from time series' inherent characteristics and hence doesn't fluctuate violently, while the local pattern is caused by some accidental factors such as sudden changes in the weather, so it's not long-lasting. As aforementioned, the global pattern should be grasped from entire training set series, and the local pattern can be captured from lookback window series. Based on this, we design *Dateformer*, it distills training set's global information into date-representations, and then leverages the information to produce a ***global prediction***. The lookback window is used to provide local pattern information, and then contributes a ***local prediction***. *Dateformer* fuses the 2 predictions into a *final prediction*. To enhance Transformer's throughput, we split time series into patches by day, and apply *Dateformer* in a patch-wise processing style. Therefore, we deal with time series forecasting problems at the day-level—predicting next $H$ days time series from historical series.

### 3.1  GLOBAL PREDICTION

We distill the whole training set's global information and store it in date-representations. Then, during inference, we can draw a *global prediction* from therein to represent learned global pattern.

$$
\begin{aligned}
Date\ Representation &= \{\boldsymbol{d}\} \qquad Positional\ Encodings = \{1,2,3,\cdots,G\} \\
\{\boldsymbol{t}_1,\boldsymbol{t}_2,\boldsymbol{t}_3,\cdots,\boldsymbol{t}_G\} &= \mathrm{Duplicate}(\boldsymbol{d}) + Positional\ Encodings \\
&= \underbrace{\{\boldsymbol{d},\boldsymbol{d},\boldsymbol{d},\cdots,\boldsymbol{d}\}}_{G\ copies} + \{1,2,3,\cdots,G\} \\
Global\ Prediction &= \mathrm{FFN}(\boldsymbol{t}_1,\boldsymbol{t}_2,\boldsymbol{t}_3,\cdots,\boldsymbol{t}_G)
\end{aligned}
\tag{2}
$$

**Predictive Network**  As equations 2, given a day's (patch's) dynamic date-representation $\boldsymbol{d}$, we duplicate it to $G$ copies ($G$ denotes series time-steps number every day), and add sequential positional encodings[4] to these copies, thereby obtaining finer time-representations $\{\boldsymbol{t}_1,\boldsymbol{t}_2,\cdots,\boldsymbol{t}_G\}$ to represent each series time-step of the day. Then, to get the day's *global prediction*, we employ a position-wise feed-forward network to map these time-representations into time series.

---

[4]We use the canonical Positional Encoding proposed by Vaswani et al. (2017), other PEs are also practicable.

**Distilling** Initially, the containers of these time-representations are empty and can't represent time series' global pattern. So, before formally training the entire Dateformer, we separately train the global predictive network on training set, to distill therein global information thereby preserving the whole training set time series' global pattern. Each day's time series patch in training set is a sample. Time features are sufficiently general for the whole historical series, so they won't distill local pattern's ad-hoc information. For some datasets, end-to-end training Dateformer is also workable.[5]

## 3.2 LOCAL PREDICTION

The local pattern information is volatile and hence can merely be available from recent observations. To better capture the local pattern from lookback window, we eliminate the learned global pattern from lookback window series. As described in the previous section, for a day in lookback window, we can get the day's *global prediction* from its date-representation. Then, the day's *Series Residual* $r$ carried only local pattern information will be produced by:

$$Series\ Residual\ \boldsymbol{r} = Series - Global\ Prediction \tag{3}$$

where $Series$ denote the day's ground-truth time series. We apply the operation to all days in lookback window, to produce the $Series\ Residuals\ \{\boldsymbol{r}_1, \boldsymbol{r}_2, \cdots\}$ of the whole lookback window. Subsequently, we utilize a vanilla Transformer to learn a *local prediction* from these $Series\ Residuals$.

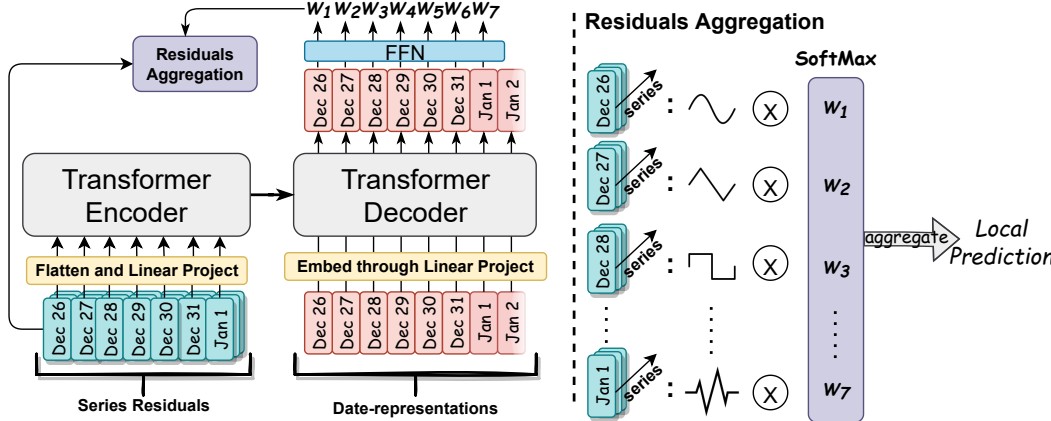

Figure 3: Take Jan 2 as an example: to get the day's *local prediction*, we utilize Transformer to learn series residuals similarities $W_i$ between Jan 2 and the lookback window (left), and use the similarities to aggregate $Series\ Residuals$ of lookback window into the *local prediction* (right).

Concretely, as shown in Figure 3, we feed the lookback window $Series\ Residuals$ into Transformer encoder. For multivariate time series, pre-flattening series is required. The encoder output will be sent into Transformer decoder as a cross information to help decoder refine input date-representations. To learn pair-wise similarities of the forecast day and each day in lookback window, the decoder eats date-representations of lookback window and forecast day, to exchange information between them. Given a lookback window of $P$ days, the similarities are calculated by:

$$Series\ Residuals = \{\boldsymbol{r}_1, \boldsymbol{r}_2, \cdots, \boldsymbol{r}_P\}$$
$$Date\ Representations = \{\boldsymbol{d}_1, \boldsymbol{d}_2, \cdots, \boldsymbol{d}_P, \boldsymbol{d}_{P+1}\}$$
$$\{\widehat{\boldsymbol{d}_1}, \widehat{\boldsymbol{d}_2}, \cdots, \widehat{\boldsymbol{d}_P}, \widehat{\boldsymbol{d}_{P+1}}\} = \text{Decoder}(Date\ Representations, \tag{4}$$
$$\text{Encoder}(Series\ Residuals))$$
$$Similarities = \{W_1, W_2, \cdots, W_P\} = \text{SoftMax}(\text{FFN}(\widehat{\boldsymbol{d}_1}, \widehat{\boldsymbol{d}_2}, \cdots, \widehat{\boldsymbol{d}_P}))$$

then, we can get a *local prediction* corresponding $\boldsymbol{d}_{P+1}$ by Aggregating $Series\ Residuals$:

$$Local\ Prediction = \text{Aggregate}(Series\ Residuals)$$
$$= W_1 \times \boldsymbol{r}_1 + W_2 \times \boldsymbol{r}_2 + \cdots + W_P \times \boldsymbol{r}_P \tag{5}$$

---

[5]We discuss it in Appendix F

### 3.3 FINAL PREDICTION

Now, we can obtain a *final prediction* by adding the *global prediction* and *local prediction*:

$$Final\ Prediction = Global\ Prediction + Local\ Prediction \tag{6}$$

For multi-day forecast, we need to encode multi-day date-representations and produce their corresponding *global predictions* and *local predictions*. Thus, we design Dateformer to automatically conduct these procedures and fuse them into the *final predictions*, just like a scheduler.

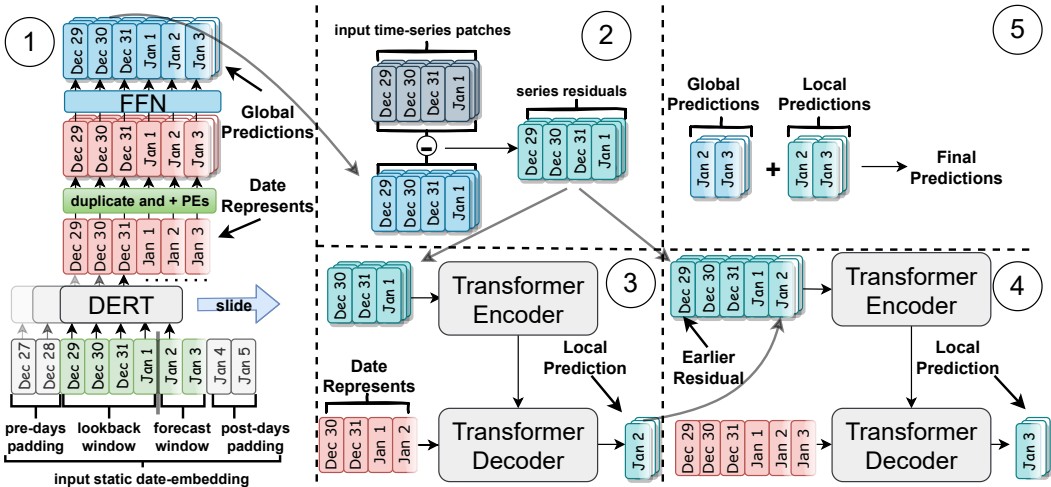

Figure 4: Work flow of Dateformer (4-predict-2 days case), 2 pre(post)-days padding. ①: sliding DERT to encode multi-day date-representations and building their *global predictions* by a single feed-forward propagation; ②: making multi-day series residuals of lookback window; ③,④: Autoregressive *local predictions*; ⑤: fusing the *global* and *local predictions* into the *final predictions*.

Referring to Figure 4, for multi-day forecast, we input into Dateformer: i) static date-embeddings of: pre-days padding, lookback window, forecast window, and post-days padding to encode multi-day date-representations; ii) lookback window series to provide local information supporting the *local prediction*. Dateformer slides DERT to encode dynamic date-representations of lookback and forecast window. Then, the *global predictions* can be produced by a single feed-forward propagation.

For the *local predictions*, Dateformer recursively calls the Transformer to day-wise predict. Autoregression is applied in the procedure—the previous prediction is used for subsequent predictions:

$$Series\ Residuals = \{\boldsymbol{r}_1, \boldsymbol{r}_2, \cdots, \boldsymbol{r}_P\}$$
$$Local\ Prediction\ \boldsymbol{r}_{P+1} = \mathrm{Aggregate}(Series\ Residuals)$$
$$Series\ Residuals = \{r_0, \boldsymbol{r}_1, \boldsymbol{r}_2, \cdots, \boldsymbol{r}_P, \boldsymbol{r}_{P+1}\} \tag{7}$$
$$Local\ Prediction\ \boldsymbol{r}_{P+2} = \mathrm{Aggregate}(Series\ Residuals)\ \cdots$$

To retard error accumulation, we adopt 3 tricks: 1) Dateformer employs Autoregression for only *local predictions*, the *global predictions* with most numerical scale of time series are generated in a single feed-forward propagation, which restricts the error's scale; 2) Dateformer regards day as time series basic unit. For fine grained time series, the day-wise Autoregression remarkable reduces errors propagation times and accelerates prediction; 3) During *local prediction* Autoregression, when a day predicted *local prediction* patch (such as $\boldsymbol{r}_{P+1}$ in equation 7) is appended to the $Series\ Residuals$ tail, Dateformer will insert earlier past a day ground-truth residual (such as $r_0$ in equation 7) into their head to balance the proportion between true local information and predicted residuals.

We didn't adopt the one-step forward generative style prediction that is proposed by Zhou et al. (2021) and followed by other Transformer-based forecast models because the style lacks scalability. For different length forecast demands, they have to re-train models. We are committed to providing a scalable forecast style to tackle forecast demands of various lengths. Although has defects of slow

prediction speed and error accumulation, Autoregression can achieve the idea. So, we still adopt Autoregression and fix its defects. Dateformer has excellent scalability. It can be trained on a short-term forecasting setup, while robustly generalized to long-term forecasting tasks. For each dataset, **we only train Dateformer once to deal with forecast demands for any number of days**.

## 4 EXPERIMENTS

**Datasets** We extensively perform experiments on 7 real-world datasets, including energy, traffic, economics, and weather: (1) *ETT* (Zhou et al., 2021) dataset collects 2 years (July 2016 to July 2018) electricity data from 2 transformers located in China, including oil temperature and load recorded every 15 minutes or hourly; (2) *ECL*[6] dataset collects the hourly electricity consumption data of 321 Portugal clients from 2012 to 2015; *PL*[7] dataset contains power load series of 2 areas in China from 2009 to 2015. It's recorded every 15 minutes and carries incomplete weather data. We eliminate the climate information and stack the 2 series into a multivariate time-series; (4) *Traffic*[8] contains hourly record value of road occupancy rate in San Francisco from 2015 to 2016; (5) *Weather*[9] is a collection of 21 meteorological indicators series recorded every 10 minutes by a German station for the 2020 whole year; (6) *ER*[10] collects the daily exchange rate of 8 countries from March 2001 to March 2022. We split all datasets into training, validation, and test set by the ratio of 6:2:2 for *ETT* and *PL* datasets and 7:1:2 for others, just following the standard protocol.

**Baselines** We select 6 strong baseline models for multivariate forecast comparisons, including 4 state-of-the-art Transformer-based, 1 RNN-based, and 1 CNN-based models: FEDformer(Zhou et al., 2022) (ICML 2022), Autoformer (Wu et al., 2021) (NeurIPS 2021), Informer (Zhou et al., 2021) (AAAI 2021 Best Paper Award), Pyraformer (Liu et al., 2021) (ICLR 2022 Oral), LSTM (Hochreiter & Schmidhuber, 1997), and TCN (Bai et al., 2018).

**Implementation details** The details of our models and Transformer-based baseline models are in AppendixI. Our code will be released after the paper's acceptance.

### 4.1 MAIN RESULTS

We show Dateformer's performance here. Due to splitting time series by day, we evaluate models with a wide range of forecast days: 1d, 3d, 7d, 30d, 90d (or 60d), and 180d covering short-, medium-, and long-term forecast demands. For coarse-grained *ER* series, we still follow the forecast horizon setups proposed by Wu et al. (2021). Our idea is to break the information bottleneck for models, so we didn't restrict models' input. Any input length is allowed as long as the model can afford it. For these baselines, if existing, we'll choose the input length recommended in their paper. Otherwise, an estimated input length will be given by us. We empirically select 7d, 9d, 12d, 26d, 46d (or 39d), and 60d as corresponding lookback days for Dateformer, and **we only train it once on the 7d-predict-1d task** to test all setups for each dataset. For fairness, all these sequence-modeling baselines are trained time-step-wise but tested day-wise. Due to the space limitation, only multivariate comparisons is shown here, see Appendix D for univariate comparisons and Table 12 for standard deviations results.

**Multivariate results** Our proposed Dateformer achieves consistent state-of-the-art performance under all forecast setups on all 7 benchmark datasets. The longer forecast horizon, the more significant improvement. For the long-term forecasting setting ($>$60 days), Dateformer gives MSE reductions: **82.5%** ($0.585 \rightarrow 0.144$, $1.672 \rightarrow 0.176$) in *PL*, **42%** ($1.056 \rightarrow 0.690$) in *ETTm1*, **38.6%** ($0.702 \rightarrow 0.431$) in *ETTh2*, **59.8%** ($0.316 \rightarrow 0.220$, $2.231 \rightarrow 0.240$) in *ECL*, **79.4%** ($2.557 \rightarrow 0.526$) in *Traffic*, and **31.5%** in *ER*. Overall, Dateformer yields a **33.6%** averaged accuracy improvement among all setups. Compared with other models, its errors rise mostly steadily as forecast horizon grows. It means that Dateformer gives the most credible and robust long-range

---

[6]https://archive.ics.uci.edu/ml/datasets/ElectricityLoadDiagrams20112014

[7]This dataset is provided by the 9th China Society of Electrical Engineering cup competition.

[8]http://pems.dot.ca.gov/

[9]https://www.bgc-jena.mpg.de/wetter/

[10]https://fred.stlouisfed.org/categories/158

Table 1: Multivariate time series forecasting results. OOM: Out Of Memory. "-" means failing to train because validation set is too short to provide even a sample. (Number) in parentheses denotes each dataset's time-steps number every day. A lower MSE or MAE indicates a better prediction. The best results are highlighted in **bold** and the second best results are highlighted with a underline.

| Models | | **Dateformer** | | FEDformer | | Autoformer | | Informer | | Pyraformer | | LSTM | | TCN | |
|---|---|---|---|---|---|---|---|---|---|---|---|---|---|---|---|
| Metric | | MSE | MAE | MSE | MAE | MSE | MAE | MSE | MAE | MSE | MAE | MSE | MAE | MSE | MAE |
| PL(96) | 1 | **0.042** | 0.141 | 0.105 | 0.231 | 0.098 | 0.201 | 0.067 | 0.157 | 0.052 | **0.139** | 0.796 | 0.577 | 0.111 | 0.211 |
| | 3 | **0.076** | **0.187** | 0.149 | 0.264 | 0.405 | 0.479 | 0.251 | 0.331 | 0.134 | 0.230 | 0.672 | 0.553 | 0.197 | 0.296 |
| | 7 | **0.093** | **0.211** | 0.196 | 0.303 | 0.398 | 0.462 | 0.370 | 0.424 | 0.211 | 0.313 | 0.970 | 0.698 | 0.216 | 0.304 |
| | 30 | **0.115** | **0.249** | 0.423 | 0.471 | 0.902 | 0.727 | 0.373 | 0.458 | 0.268 | 0.381 | 2.025 | 1.115 | 0.698 | 0.649 |
| | 90 | **0.144** | **0.291** | OOM | OOM | OOM | OOM | OOM | OOM | 0.585 | 0.590 | 3.394 | 1.359 | 1.415 | 0.956 |
| | 180 | **0.176** | **0.320** | OOM | OOM | OOM | OOM | OOM | OOM | OOM | OOM | 3.312 | 1.462 | 1.672 | 1.004 |
| ETTm1(96) | 1 | **0.322** | **0.355** | 0.341 | 0.390 | 0.491 | 0.476 | 0.640 | 0.570 | 0.515 | 0.505 | 1.058 | 0.709 | 0.734 | 0.667 |
| | 3 | **0.368** | **0.381** | 0.419 | 0.439 | 0.598 | 0.522 | 0.963 | 0.750 | 0.849 | 0.713 | 1.505 | 0.890 | 0.833 | 0.722 |
| | 7 | **0.417** | **0.410** | 0.473 | 0.468 | 0.646 | 0.547 | 1.129 | 0.800 | 1.053 | 0.818 | 1.897 | 1.028 | 0.881 | 0.729 |
| | 30 | **0.438** | **0.446** | 0.547 | 0.525 | 0.681 | 0.581 | 1.132 | 0.831 | 1.020 | 0.806 | 2.970 | 1.240 | 1.033 | 0.811 |
| | 90 | **0.690** | **0.600** | OOM | OOM | OOM | OOM | OOM | OOM | 1.056 | 0.845 | 1.855 | 1.010 | 1.081 | 0.821 |
| ETTh2(24) | 1 | **0.234** | **0.306** | 0.246 | 0.327 | 0.289 | 0.364 | 1.606 | 0.997 | 0.412 | 0.498 | 1.074 | 0.723 | 1.126 | 0.874 |
| | 3 | **0.311** | **0.363** | 0.334 | 0.381 | 0.347 | 0.395 | 1.928 | 1.111 | 1.140 | 0.832 | 1.583 | 0.873 | 1.900 | 1.138 |
| | 7 | **0.383** | **0.413** | 0.412 | 0.426 | 0.451 | 0.451 | 6.200 | 2.024 | 4.877 | 1.747 | 2.429 | 1.111 | 4.410 | 1.741 |
| | 30 | **0.437** | **0.472** | 0.466 | 0.483 | 0.510 | 0.511 | 4.091 | 1.717 | 4.674 | 1.869 | 2.014 | 1.052 | 2.919 | 1.503 |
| | 90 | **0.431** | **0.486** | 0.719 | 0.618 | 0.702 | 0.631 | 2.571 | 1.217 | 3.330 | 1.511 | 3.754 | 1.436 | 3.356 | 1.503 |
| ECL(24) | 1 | **0.113** | **0.218** | 0.169 | 0.288 | 0.174 | 0.293 | 0.328 | 0.412 | 0.268 | 0.372 | 0.368 | 0.425 | 0.365 | 0.436 |
| | 3 | **0.148** | **0.251** | 0.186 | 0.302 | 0.215 | 0.329 | 0.358 | 0.430 | 0.308 | 0.388 | 0.475 | 0.474 | 0.434 | 0.471 |
| | 7 | **0.163** | **0.266** | 0.201 | 0.316 | 0.208 | 0.320 | 0.387 | 0.459 | 0.281 | 0.381 | 0.786 | 0.634 | 0.362 | 0.432 |
| | 30 | **0.187** | **0.291** | 0.242 | 0.351 | 0.259 | 0.364 | 0.400 | 0.460 | 0.288 | 0.382 | 1.304 | 0.834 | 0.442 | 0.484 |
| | 90 | **0.220** | **0.322** | 0.316 | 0.404 | 0.382 | 0.439 | 0.550 | 0.552 | OOM | OOM | 1.815 | 1.048 | 0.501 | 0.513 |
| | 180 | **0.240** | **0.339** | - | - | - | - | - | - | - | - | 2.231 | 1.159 | - | - |
| Traffic(24) | 1 | **0.343** | **0.252** | 0.547 | 0.357 | 0.554 | 0.363 | 0.678 | 0.383 | 0.600 | 0.337 | 1.220 | 0.622 | 0.814 | 0.480 |
| | 3 | **0.430** | **0.284** | 0.581 | 0.367 | 0.625 | 0.391 | 0.721 | 0.404 | 0.635 | 0.358 | 1.902 | 0.878 | 1.582 | 0.743 |
| | 7 | **0.434** | **0.289** | 0.613 | 0.382 | 0.705 | 0.439 | 0.768 | 0.425 | 0.639 | 0.356 | 2.509 | 1.114 | 0.803 | 0.444 |
| | 30 | **0.469** | **0.308** | 0.652 | 0.395 | 0.686 | 0.420 | 0.959 | 0.534 | OOM | OOM | 2.413 | 1.089 | 1.120 | 0.615 |
| | 90 | **0.526** | **0.339** | - | - | - | - | - | - | - | - | 2.557 | 1.128 | - | - |
| Weather(144) | 1 | **0.220** | **0.288** | 0.234 | 0.304 | 0.327 | 0.377 | 0.370 | 0.424 | 0.231 | 0.310 | 5.868 | 1.478 | 0.267 | 0.328 |
| | 3 | **0.281** | **0.329** | 0.338 | 0.375 | 0.370 | 0.402 | 0.628 | 0.560 | 0.388 | 0.427 | 4.232 | 1.417 | 0.468 | 0.452 |
| | 7 | **0.320** | **0.360** | 0.495 | 0.472 | 0.474 | 0.461 | 1.093 | 0.768 | 0.442 | 0.460 | 5.727 | 1.469 | 0.523 | 0.510 |
| | 30 | **0.414** | **0.428** | 0.688 | 0.580 | 0.724 | 0.604 | 2.789 | 1.303 | 0.823 | 0.676 | 7.141 | 1.804 | 0.878 | 0.711 |
| | 60 | **0.539** | **0.531** | - | - | - | - | - | - | - | - | 8.168 | 1.766 | - | - |
| ER(1) | 96 | **0.022** | **0.107** | 0.041 | 0.154 | 0.040 | 0.154 | 0.248 | 0.361 | 0.194 | 0.340 | 0.452 | 0.520 | 0.482 | 0.541 |
| | 192 | **0.043** | **0.152** | 0.062 | 0.194 | 0.061 | 0.193 | 0.430 | 0.474 | 0.337 | 0.440 | 0.536 | 0.558 | 0.496 | 0.558 |
| | 336 | **0.070** | **0.195** | 0.087 | 0.233 | 0.096 | 0.246 | 0.756 | 0.642 | 0.607 | 0.600 | 0.825 | 0.705 | 0.557 | 0.586 |
| | 720 | **0.112** | **0.255** | 0.165 | 0.317 | 0.389 | 0.422 | 1.073 | 0.773 | 0.963 | 0.772 | 0.866 | 0.729 | 0.677 | 0.652 |

forecast. Note that our Dateformer still contributes the best predictions on the *ER* series which is **coarse-grained time series without obvious periodicity.**

## 4.2 ANALYSIS

We try to analyze the 2 forecast components' contributions to the *final prediction* and explain why Dateformer's predictive errors rise so slow as forecast days extend. We use separate *global prediction* or *local prediction* component to predict and compare their results, as shown below.

Table 2: Multivariate time series forecasting comparison of different forecast components results. The results highlighted in **bold** indicate which component contributes the best prediction.

| Datasets | | *PL* | | | *ECL* | | | *Traffic* | | |
|---|---|---|---|---|---|---|---|---|---|---|
| Forecast Days | | 1 | 30 | 180 | 1 | 30 | 180 | 1 | 7 | 90 |
| Global | MSE | 0.129 | 0.132 | **0.156** | 0.310 | 0.303 | 0.302 | 0.634 | 0.640 | 0.656 |
| Prediction | MAE | 0.264 | 0.268 | **0.297** | 0.395 | 0.392 | 0.392 | 0.355 | 0.356 | 0.362 |
| Local | MSE | **0.029** | 0.406 | 1.386 | **0.101** | 0.249 | 0.378 | **0.330** | 0.545 | 0.725 |
| Prediction | MAE | **0.115** | 0.457 | 0.908 | **0.201** | 0.327 | 0.412 | **0.239** | 0.341 | 0.414 |
| Final | MSE | 0.042 | **0.115** | 0.176 | 0.113 | **0.187** | **0.240** | 0.343 | **0.434** | **0.526** |
| Prediction | MAE | 0.141 | **0.249** | 0.320 | 0.218 | **0.291** | **0.339** | 0.252 | **0.289** | **0.339** |

It can be seen that errors of the separate *global prediction* hardly rise as forecast horizon grows. The separate *local prediction*, however, is rapidly deteriorating. This may prove our hypothesis that time series have 2 patterns of components: global and local pattern. As stated in section 3, the global pattern will not fluctuate violently, while the local pattern is not long-lasting. Therefore, as forecast horizon grows, *global prediction* that still keep stable errors is especially important for long-term forecasting. Although do best in short-term forecasting, errors of *local prediction* increase rapidly as forecast days extend because local pattern is not long-lasting. As time goes on, the local pattern shifts gradually. For distant future days, current local pattern even degenerates into a noise that encumbers predictions: comparing the 180 days prediction cases on *PL*, the best prediction is contributed by the separate *global prediction* instead of the *final prediction* that disturbed by the noise. Not only local pattern, but Dateformer also grasps global pattern, so its errors rise mostly steadily. To intuitively understand the 2 patterns' contributions, we use Auto-correlation that presents the same periodicity with source series to analyze the seasonality and trend of *ETT* oil temperature series.

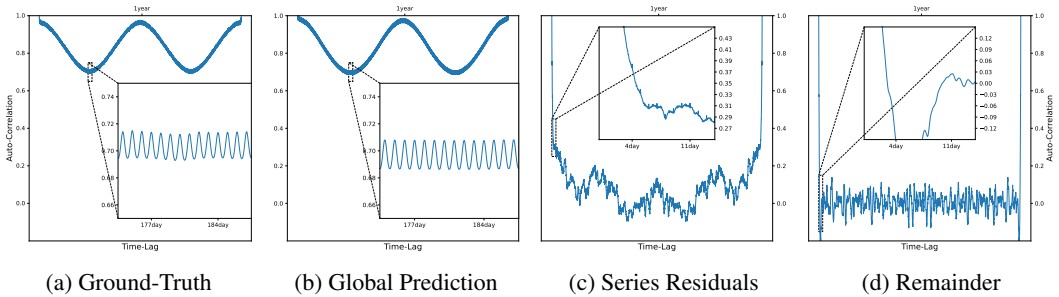

|                (a) Ground-Truth                |              (b) Global Prediction               |              (c) Series Residuals               |                (d) Remainder                 |

Figure 5: Auto-correlation of four time series from *ETT* oil temperature series: (a) Ground-truth; (b) Global Prediction; (c) Ground-truth − Global Prediction; (d) Ground-truth − Final Prediction.

As shown in Figure 5a, the *ETT* oil temperature series mixes complex multi-scale seasonality, and the *global prediction* almost captures it perfectly in 5b. But see Figure 5c, we can find *global prediction* fail to accurately estimates series mean: although a sharp descent occurs in the left end after eliminate the *global prediction*, it didn't immediately drop to near 0. Because local pattern will effect series trends. The local pattern is not long-lasting, so we can only approximate it from lookback window. After subtracting the *final prediction*, we get an Auto-correlation series of random oscillations around 0 in 5d, which indicates source series is as unpredictable as white noise series.

### 4.3 ABLATION STUDIES

We conduct ablation studies about input lookback window length and proposed time-representing method, related results and discussions are in Appendix E. Due to the space limitation, we only list major finds here. (1) Our Dateformer can benefit from longer lookback windows, but not for other Transformers; (2) The proposed dynamic time-representing method can effectively enhance Dateformer's performance; (3) But sequence-modeling Transformers, which underestimate the significance of time and let it just play an auxiliary role, may not benefit from better time-encodings.

In addition, we also conduct ablation studies about 2 pre-training tasks that are presented in section 2. Beyond all doubt, the pre-training task of mean prediction is naturally effective. Removing this task to pre-train, Dateformer's predictions remarkable deteriorates. But for some datasets, pre-training DERT on only this task also damages Dateformer's performance. Auto-correlation prediction can induce DERT to concern time series intraday information, thereby encoding finer date-representations.

## 5 CONCLUSIONS

In this paper, we challenge employing merely vanilla Transformers to predict long-term series. We analyze information characteristic of time series and propose splitting time series into patches to enhance Transformer's throughout. Besides, we distill the whole training set's global information and store it in time representations, to help the model grasp time series' global pattern. The proposed Dateformer yields consistent state-of-the-art performance on extensive 7 real-world datasets.

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

## A   FUTURE WORKS

In this paper, as the preliminary work, we briefly introduced the dynamic date-representation and DERT, but didn't talk about their transferability. Actually, under certain conditions, a DERT that is pre-trained on a time series dataset can transfer to other similar datasets to encode date-representations. For example, *PL* and *ETT* are similar electrical series datasets recorded in different cities of China, *PL* range from 2009 to 2015 while *ETT* range from 2016 to 2018, but we found the DERT that is pre-trained on *PL* can directly transfer to forecast tasks on *ETT*. More interestingly, with the transferred DERT, Dateformer demonstrates a fairly good few sample learning potential. On the *ETT* series, even though only one-third of training samples are provided, Dateformer still can converge to the same performance level as the full training samples provided (see Table 3). The details are not completely clear yet, we will further study in future works.

Table 3: Few sample learning multivariate results on *ETTm1*, only MSE is reported.

| Models | Dateformer-*ETTm1*-pre-trained | | | | Dateformer-*PL*-pre-trained | | | |
|---|---|---|---|---|---|---|---|---|
| Samples | 353 | 233 | 113 | 1 | 353 | 233 | 113 | 1 |
| 1 | 0.326 | 0.331 | 0.333 | 0.422 | **0.322** | 0.324 | 0.332 | 0.373 |
| 3 | 0.370 | 0.374 | 0.383 | 0.445 | 0.368 | **0.365** | 0.381 | 0.399 |
| 7 | 0.418 | 0.422 | 0.427 | 0.478 | 0.417 | **0.414** | 0.426 | 0.443 |
| 30 | 0.443 | 0.442 | 0.435 | 0.476 | 0.438 | 0.437 | **0.435** | 0.453 |
| 90 | 0.702 | 0.711 | 0.680 | 0.698 | 0.690 | 0.680 | **0.678** | 0.696 |

Referring to Table 3, we train 2 Dateformers on *ETT* series ranging full 12 months (353 samples), 8 months (233 samples), 4 months (113 samples) and 8 days (1 sample), to test all forecast horizon setups on the same test set. The Dateformer whose DERT is pre-trained on *PL* series shows the same performance level, even a slightly better because *PL* is a bigger dataset with a longer time span. And fewer samples are required: 4 months of *ETT* series is enough for Dateformer to convergence.

## B   RELATED WORKS

Time series forecasting is an enduring research topic, and numerous works have been developed to deal with the task. They can be roughly divided into two categories: statistical methods and deep learning models. ARIMA (Box & Jenkins, 1968), Prophet (Taylor & Letham, 2018), and the filtering methods are representative methods of the former. The deep learning models mainly include RNN-based (Recurrent Neural Network based), CNN-based (Convolutional Neural Network based), and Transformer-based structures.

LSTM (Hochreiter & Schmidhuber, 1997) and GRU (Chung et al., 2014) employ gating mechanisms to extend series dependence learning distance and relieve the gradient vanishing or explosion of RNN. DeepAR (Salinas et al., 2020) further combines LSTM and Autoregression for time series probabilistic distribution forecasting. LSTNet (Lai et al., 2018) adopts CNN and recurrent-skip connections to modeling short- and long-term patterns of time series. Some RNN-based works introduced the attention mechanism to capture long-range temporal dependence, so as to improve predictions (Qin et al., 2017). However, RNN's intrinsic flaws (difficulty capturing long-range dependence, slow reasoning, and accumulating error) prevent them from predicting long-term series.

Temporal convolutional network (TCN) (Bai et al., 2018) is CNN-based representative work, which models temporal dependence with the causal convolution. DeepGLO (Sen et al., 2019) also mentioned the concept of global and local, and employs TCN to model them. Nevertheless, **the global concept in their paper refers to relationships between related other time series, which is distinct from this paper refers to the global historical observation on the series itself.**

Transformer (Vaswani et al., 2017) recently becomes popular in long-term series forecasting, owing to its excellent long-range dependence capturing capability. But directly applying Transformer to long-term series forecasting is computationally prohibitive due to inside self-attention's quadratic complexity about sequence length in both memory and time. Many studies are proposed to resolve the problem. LogTrans (Li et al., 2019) presented *LogSparse* attention that reduces the

computational complexity to $\mathcal{O}(L(\log L)^2)$, and Reformer (Kitaev et al., 2020) presented local-sensitive hashing (LSH) attention with $\mathcal{O}(L\log L)$ complexity. Informer (Zhou et al., 2021) proposed *probSparse* attention of $\mathcal{O}(L\log L)$ complexity, and further renovated the vanilla Transformer architecture. Autoformer (Wu et al., 2021) and FEDformer (Zhou et al., 2022) built decomposed blocks in Transformer and introduced low complexity enhanced attention in frequency domain or Auto-correlation to replace self-attention. Pyraformer (Liu et al., 2021) adopted hierarchical multi-resolution PAM attention to achieve $\mathcal{O}(L)$ complexity and learn multi-scale temporal dependence. To extend forecast horizon, all these variant Transformers designed various modified self-attentions or substitutes to reduce computational complexity, so as to tackle longer time series.

In CV community, ViT (Dosovitskiy et al., 2020) split images into patches, thereby enabling vanilla Transformers to tackle abundant pixels. Compared to highly abstract human language, we argue that information characteristic of time series is more similar to it of images. They both are natural signals and exist numerical continuity between adjacent signals. Inspired by this, we follow ViT to split time series into patches, thereby enhancing time series forecasting Transformers' throughput, and enabling vanilla Transformers to predict long-term series.

In addition to these structures, there are also other neural network methods such as N-BEATS (Oreshkin et al., 2019) that uses decomposition. Above most forecast methods can be summarized as learning a mapping from lookback window to forecast window, and they some leverage time feature assist prediction. However, in the end, they are confined by information because they can only rely on the local information in lookback window. To our best knowledge, we are the first time series forecasting work that distills the whole training set's global information into time-representations and takes time instead of series as the primary modeling entities—time-modeling strategy.

## C  DATE EMBEDDING

Observing the movement of celestial planets, changes in temperature, or the rise and fall of plants, our forefathers distilled a set of laws, that is calendars. A wealth of wisdom is encapsulated in the calendar. People's activities are guided by the calendar and we believe that's the fundamental source of seasonality in many time series. We try to introduce the wisdom in our models as a priori knowledge. We use the Gregorian calendar as solar calendar and the traditional Chinese calendar as lunar calendar to deduce our date-embedding. Besides, the vacation and weekday information is also taken into count. In our code, we provide the date-embeddings range from 2001 to 2023 for 12 countries or regions: Australia, British, Canada, China, Germany, Japan, New Zealand, Portugal, Singapore, Switzerland, USA, and San Francisco.

Our static date-embedding stacks the following date features: *abs_day, year, day (month_day), year_day, weekofyear, lunar_year, lunar_month, lunar_day, lunar_year_day, dayofyear, dayofmonth, monthofyear, dayofweek, dayoflunaryear, dayoflunarmonth, monthoflunaryear, jieqiofyear, jieqi_day, dayofjieqi, holidays, workdays, residual_holiday, residual_workday.* As an example, we embed today by following equations:

$$abs\_day = \frac{\text{days that have passed from December 31, 2000}}{365.25 * 5}$$

$$year = \frac{\text{this year} - 1998.5}{25}$$

$$day = \frac{\text{days that have passed in this month}}{31}$$

$$year\_day = \frac{\text{days that have passed in this year}}{366}$$

$$weekofyear = \frac{\text{weeks that have passed in this year}}{54}$$

$$lunar\_year = \frac{\text{this lunar year} - 1998.5}{25}$$

$$lunar\_month = \frac{\text{this lunar month}}{12}$$

$$lunar\_day = \frac{\text{days that have passed in this lunar month}}{30}$$

$$lunar\_year\_day = \frac{\text{days that have passed in this lunar year}}{384}$$

$$dayofyear = \frac{\text{days that have passed in this year} - 1}{\text{total days in this year} - 1} - 0.5$$

$$dayofmonth = \frac{\text{days that have passed in this month} - 1}{\text{total days in this month} - 1} - 0.5$$

$$monthofyear = \frac{\text{months that have passed in this year} - 1}{11} - 0.5$$

$$dayofweek = \frac{\text{days that have passed in this week} - 1}{6} - 0.5$$

$$dayoflunaryear = \frac{\text{days that have passed in this lunar year} - 1}{\text{total days in this lunar year} - 1} - 0.5$$

$$dayoflunarmonth = \frac{\text{days that have passed in this lunar month} - 1}{\text{total days in this lunar month} - 1} - 0.5$$

$$monthoflunaryear = \frac{\text{lunar months that have passed in this lunar year} - 1}{11} - 0.5$$

$$jieqiofyear = \frac{\text{solar terms that have passed in this year} - 1}{23} - 0.5$$

$$jieqi\_day = \frac{\text{days that have passed in this solar term}}{15}$$

$$dayofjieqi = \frac{\text{days that have passed in this solar term} - 1}{\text{total days in this solar term} - 1} - 0.5$$

We standardize these date features by respective corresponding maximum. For unbounded features (e.g., $abs\_day$ and $year$), we use a large number to standardize them. For example, we can use 100 to standardize $year$, then need not worry about it in our lifetime. With DERT encoding, our model shows excellent generalization when facing a date never seen before (See Table 2). Because too complicated, some date features' equations are omitted. Email authors if interested.

# D    UNIVARIATE RESULTS

**Baselines**    We also select 6 strong baseline models for univariate forecast comparisons, covering state-of-the-art deep learning models and classic statistical methods: FEDformer(Zhou et al., 2022), Autoformer (Wu et al., 2021), Informer (Zhou et al., 2021), N-BEATS (Oreshkin et al., 2019), DeepAR (Salinas et al., 2020), and ARIMA (Box & Jenkins, 1968).

Table 4: Univariate series forecast results. A lower MSE or MAE indicates a better prediction. The best results are highlighted in **bold** and the second best results are highlighted with a underline.

| Models | | Dateformer | | FEDformer | | Autoformer | | Informer | | N-BEATS | | DeepAR | | ARIMA | |
|---|---|---|---|---|---|---|---|---|---|---|---|---|---|---|---|
| Metric | | MSE | MAE | MSE | MAE | MSE | MAE | MSE | MAE | MSE | MAE | MSE | MAE | MSE | MAE |
| PL(96) | 1 | **0.040** | **0.133** | 0.132 | 0.249 | 0.145 | 0.240 | 0.058 | 0.144 | 0.092 | 0.170 | 0.211 | 0.272 | 0.996 | 0.801 |
| | 3 | **0.087** | **0.190** | 0.183 | 0.290 | 0.350 | 0.416 | 0.197 | 0.277 | 0.209 | 0.272 | 0.543 | 0.506 | 1.163 | 0.878 |
| | 7 | **0.103** | **0.212** | 0.236 | 0.326 | 0.491 | 0.504 | 0.371 | 0.399 | 0.242 | 0.295 | 0.554 | 0.512 | 1.325 | 0.926 |
| | 30 | **0.113** | **0.239** | 0.516 | 0.507 | 0.841 | 0.696 | 0.418 | 0.470 | 0.430 | 0.448 | 1.758 | 1.063 | 5.594 | 1.292 |
| | 90 | **0.135** | **0.276** | OOM | OOM | OOM | OOM | OOM | OOM | 0.442 | 0.517 | 1.311 | 0.863 | 7.801 | 1.739 |
| | 180 | **0.162** | **0.305** | OOM | OOM | OOM | OOM | OOM | OOM | OOM | OOM | 1.932 | 1.115 | 9.824 | 2.039 |
| Traffic(24) | 1 | **0.097** | **0.167** | 0.181 | 0.288 | 0.257 | 0.361 | 0.211 | 0.304 | 0.139 | 0.229 | 0.326 | 0.399 | 0.461 | 0.499 |
| | 3 | **0.118** | **0.199** | 0.231 | 0.342 | 0.293 | 0.390 | 0.232 | 0.327 | 0.144 | 0.236 | 0.441 | 0.457 | 0.796 | 0.703 |
| | 7 | **0.122** | **0.204** | 0.228 | 0.329 | 0.343 | 0.425 | 0.250 | 0.342 | 0.162 | 0.260 | 0.839 | 0.683 | 1.240 | 0.915 |
| | 30 | **0.134** | **0.221** | 0.252 | 0.349 | 0.294 | 0.389 | 0.302 | 0.393 | 0.200 | 0.306 | 0.518 | 0.503 | 1.627 | 1.086 |
| | 90 | **0.157** | **0.236** | - | - | - | - | - | - | - | - | 0.695 | 0.617 | 1.739 | 1.130 |
| ER(1) | 96 | **0.042** | **0.157** | 0.069 | 0.206 | 0.097 | 0.238 | 0.341 | 0.508 | 0.235 | 0.378 | 0.179 | 0.342 | 0.086 | 0.190 |
| | 192 | **0.079** | **0.223** | 0.108 | 0.266 | 0.124 | 0.281 | 0.453 | 0.581 | 1.697 | 1.112 | 0.427 | 0.584 | 0.213 | 0.282 |
| | 336 | **0.118** | **0.267** | 0.149 | 0.312 | 0.144 | 0.305 | 0.695 | 0.738 | 5.616 | 2.119 | 0.535 | 0.659 | 0.191 | 0.320 |
| | 720 | **0.187** | **0.344** | 0.209 | 0.373 | 0.297 | 0.439 | 2.540 | 1.522 | 5.154 | 1.767 | 1.221 | 1.012 | 0.267 | 0.364 |

**Univariate results** In the univariate time series forecasting setting, Dateformer still achieves consistent state-of-the-art performance under all forecast setups. For the long-term forecasting setting, Dateformer gives MSE reduction: **82.3%** ($0.442 \rightarrow 0.135$, $1.932 \rightarrow 0.162$) in *PL*, **77.4%** ($0.695 \rightarrow 0.157$) in *Traffic*, **23.7%** in *ER*. Overall, Dateformer yields a **41.6%** averaged accuracy improvement among all univariate forecast setups on the given 3 representative datasets.

# E  ABLATION STUDIES

## E.1  IMPACT OF INPUT LENGTH

In this section, we study the impact on several Transformer-based models of different input lengths. On representative 8 forecast tasks, which cover short-, middle-, and long-term forecasting on various time series datasets, we gradually extend their lookback window and record their predictive errors of different input window sizes. The results are shown in the form of line charts as follows.

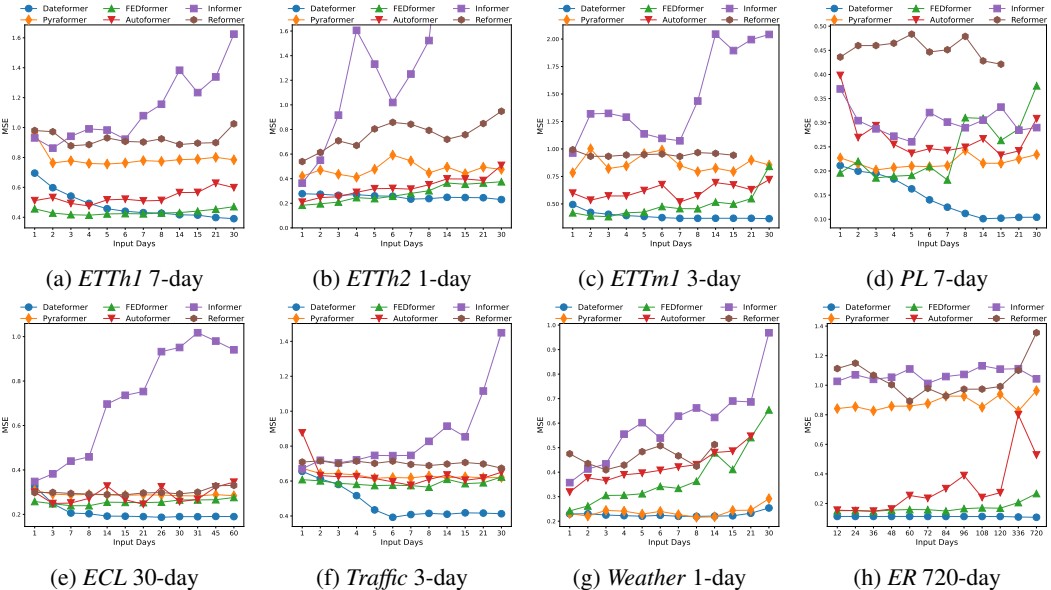

(a) *ETTh1* 7-day  (b) *ETTh2* 1-day  (c) *ETTm1* 3-day  (d) *PL* 7-day

(e) *ECL* 30-day  (f) *Traffic* 3-day  (g) *Weather* 1-day  (h) *ER* 720-day

Figure 6: The errors lines of several Transformers with different input lookback days of: predicting next (a) 7-day on *ETTh1*,(b) 1-day on *ETTh2*,(c) 3-day on *ETTm1*,(d) 7-day on *PL*,(e) 30-day on *ECL*,(f) 3-day on *Traffic*,(g) 1-day on *Weather*,(h) 720-day on *ER* time series.

As shown in Figure 6, with lookback window extends, Dateformer's predictive errors gradually reduce until stable, which indicates that Dateformer can leverage more information to continuously improve prediction. Although not the best under narrow lookback windows, Dateformer finally outperforms other all baseline Transformers as lookback window grows. But other Transformers may not benefit from larger input windows. Their performances are unstable, and even deteriorate as input length extends. This is against common sense: more information intake should lead to better predictions. Compared to them, the information utilization upper limit of Dateformer is higher. We can always expect better predictions by feeding longer lookback window series, at least not worse.

## E.2  IMPACT OF TIME ENCODING

In this paper, we contribute 2 time-encodings: the static date-embedding that is generated by stacking more date features straightforwardly, and the dynamic date-representation that further considers date contextual information. So, we try to clarify which time-encoding is better, and how well other Transformers perform when better time-encodings are provided.

### E.2.1 WHICH TIME ENCODING BETTER?

The better time-encoding should be able to represent time more accurately and accommodate more time series global information. Thus, we employ the 2 time-encoding to distill time series' global information through 2 *global prediction* components with roughly the same number of parameters, and then check their *global prediction* quality. The results are shown below.

Table 5: *Global prediction* multivariate comparison using 2 time-encoding.

| Datasets | | *PL* | | | *ECL* | | | *Traffic* | | |
|---|---|---|---|---|---|---|---|---|---|---|
| Forecast Days | | 1 | 30 | 180 | 1 | 30 | 180 | 1 | 7 | 90 |
| Dynamic | MSE | **0.129** | **0.132** | **0.156** | **0.310** | **0.303** | **0.302** | **0.634** | **0.640** | **0.656** |
| Representation | MAE | **0.264** | **0.268** | **0.297** | **0.395** | **0.392** | **0.392** | **0.355** | **0.356** | **0.362** |
| Static | MSE | 0.177 | 0.181 | 0.193 | 0.316 | 0.306 | 0.305 | 0.655 | 0.659 | 0.669 |
| Embedding | MAE | 0.306 | 0.310 | 0.325 | 0.398 | 0.393 | 0.393 | 0.363 | 0.362 | 0.363 |

As shown in Table 5, the *global prediction* component that employs dynamic date-representation always contributes a better *global prediction*. This is enough to prove the effectiveness of our proposed dynamic time-representing method. In order to more intuitively show the importance of date contextual information, and how DERT pays attention to date context, we visualize some attention weight distributions of DERT encoder. The figures are shown as follows.

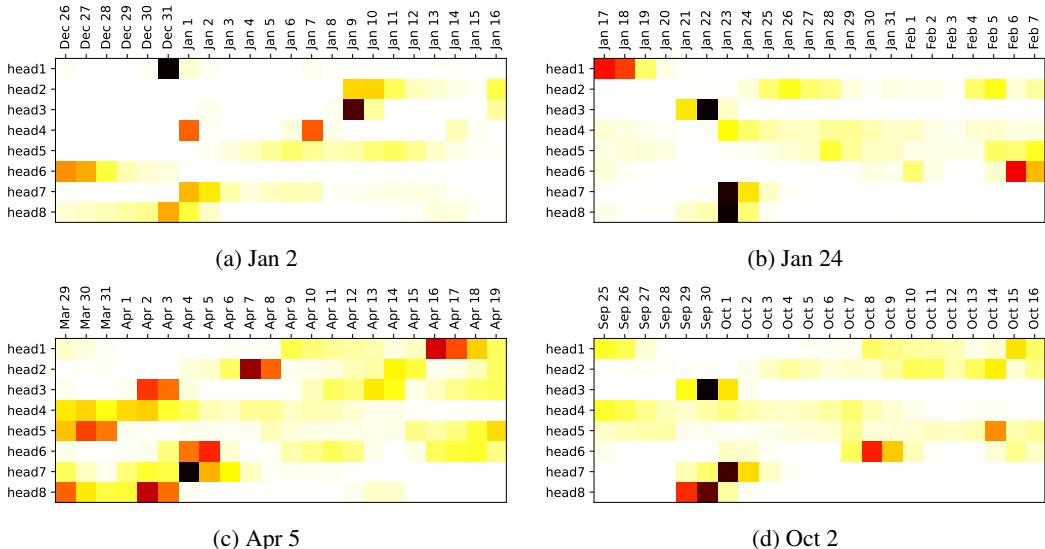

(a) Jan 2

(b) Jan 24

(c) Apr 5

(d) Oct 2

Figure 7: DERT encoder attention weight distributions from *PL* dataset during 2012, the darker the cell, the greater the attention weight. (a) is encoding Jan 2, Jan 1 is New Year's day; (b) is encoding Jan 24, in traditional Chinese lunar calendar, Jan 22 is New Year's Eve and Jan 23 is the Spring Festival; (c) is encoding Apr 5, Apr 4 is the Qingming Festival. (d) is encoding Oct 2, Sep 30 is the Mid-Autumn Festival in Chinese lunar calendar, and Oct 1 is National Day in Gregorian calendar.

In Figure 7, when approaching important festivals, DERT pays more attention to these festivals. The more important festival, the more concentrated attention. In China, the Spring Festival is the most important festival of a year. So, we can see the most intensive attention on the day in Figure 7b, and other days barely receive little attention. But referring to Figure 7c , although very intensive attention on Apr 4, some attention is still distributed to other days. Because the Qingming Festival is not as important as the Spring Festival. DERT will not pay excessive attention to it. This is completely in line with Chinese habits. Besides, DERT can also tackle the changeable date semantics in different calendars. In the traditional Chinese lunar calendar, Sep 30, 2012, is the Mid-Autumn Festival while Oct 1 is Chinese National Day in the Gregorian calendar. The interval between the 2 festivals change

every year, and in some years they are even on the same day. In Figure 7d, they both exert influence on Oct 2. We can not sure how their influence change when they are on the same day or on adjacent days, so let DERT learn from data. We also call the similar problems as date polysemy. Note that **we didn't tell DERT which day is a festival, all knowledge about festivals is learned by itself.**

### E.2.2  HOW WELL OTHER TRANSFORMERS PERFORM USING BETTER TIME ENCODINGS?

These Transformer-based baselines for comparison also leverage time features to assistant predictions. They adopt the time embedding that proposed by Zhou et al. (2021). We are inspired by it too, so follow it and stack more date features into our static date-embedding. In this section, we try to clarify how well they perform when better time-encodings are provided. We modify these Transformers to employ the time-encodings mentioned in the previous section, to compare the impact of different time-encoding on them. To better understand the role of time for these sequence-modeling Transformers, we eliminate time embedding as a comparison. The results are as follows.

Table 6: Ablation of time-encoding on *ETTh1* multivariate forecast tasks with MSE metric. **Without** eliminates time-encoding. **Origin** adopts time-encoding proposed by Zhou et al. (2021). **Static** and **Dynamic** denote the static date-embedding and dynamic date-representation respectively. Note that **Origin** and **Static** are essentially the same, **Static** just simply collects more time features.

| Forecat Days | 3 | | | | 90 | | | |
|---|---|---|---|---|---|---|---|---|
| Models | Without | Origin | Static | Dynamic | Without | Origin | Static | Dynamic |
| FEDformer | **0.359** | 0.360 | 0.374 | 0.367 | **0.853** | 1.027 | 1.250 | 1.142 |
| Autoformer | **0.415** | 0.436 | 0.441 | 0.422 | 1.029 | **0.929** | 1.327 | 1.257 |
| Informer | **0.695** | 0.788 | 0.747 | 0.872 | 1.395 | 1.291 | **1.168** | 1.672 |
| Pyraformer | 0.606 | 0.605 | 0.600 | **0.591** | 1.064 | 1.100 | 1.079 | **0.987** |
| Reformer | 0.723 | 0.776 | 0.840 | **0.699** | 1.179 | 1.210 | 1.245 | **1.085** |
| Dateformer | - | 0.386 | 0.413 | **0.369** | - | 0.721 | 0.711 | **0.691** |

Table 7: Ablation of time-encoding on *Traffic* multivariate forecast tasks with MSE metric. - means can't train, because time-encoding is necessary for Dateformer. Best results are highlighted in **bold**.

| Forecat Days | 7 | | | | 30 | | | |
|---|---|---|---|---|---|---|---|---|
| Models | Without | Origin | Static | Dynamic | Without | Origin | Static | Dynamic |
| FEDformer | 0.626 | 0.613 | 0.601 | **0.592** | 0.676 | 0.652 | 0.630 | **0.624** |
| Autoformer | 0.807 | 0.705 | 0.653 | **0.641** | 1.285 | 0.686 | 0.673 | **0.667** |
| Informer | **0.751** | 0.768 | 0.787 | 0.774 | 0.966 | **0.959** | 1.196 | 1.072 |
| Pyraformer | 0.646 | **0.639** | 0.644 | 0.671 | OOM | OOM | OOM | OOM |
| Reformer | 0.978 | 0.720 | 0.703 | **0.696** | 1.470 | **0.683** | 0.721 | 0.885 |
| Dateformer | - | 0.483 | 0.451 | **0.434** | - | 0.510 | 0.487 | **0.475** |

Referring to Table 6 and 7, as a time-modeling method that mainly models time, Dateformer consistently benefits from better time-encodings. However, better time-encodings may not enhance other Transformers that mainly model series, and they only leverage time-encoding in an auxiliary style just like positional encoding. In some forecast cases, better time-encodings can improve their predictions. But in some other cases, eliminating time-encoding results in the best prediction. Their utilization of time is unstable. Besides, simply collecting more time features does not necessarily work, sometimes even encumber predictions (see 3 days prediction case in Table 6).

### E.3 Ablation study about pre-training tasks

In section 2, we design 2 pre-training tasks for DERT: mean prediction and Auto-correlation prediction. Here, we check their effectiveness. We use the separate pre-training task to train DERT, and compare their prediction results. The results are shown as follows.

Table 8: Multivariate forecast comparison of Dateformer using different pre-training tasks. **Mean** adopts only pre-training task of mean prediction. **Auto** employs separate Auto-correlation prediction to pre-train DERT. **Both** combines the 2 pre-training tasks together, and it's adopted by us. The best results are highlighted in **bold**. "-" denotes lacking test samples to report result.

| Time Series Datasets | | *PL* | | | *ECL* | | | *ETTm1* | | |
|---|---|---|---|---|---|---|---|---|---|---|
| Forecast Days | Metric | Mean | Auto | Both | Mean | Auto | Both | Mean | Auto | Both |
| 1 | MSE | 0.044 | **0.041** | 0.042 | 0.123 | 0.128 | **0.113** | 0.328 | 0.336 | **0.326** |
| | MAE | 0.146 | **0.138** | 0.141 | 0.232 | 0.240 | **0.218** | **0.355** | 0.362 | 0.356 |
| 3 | MSE | 0.096 | 0.094 | **0.076** | 0.159 | 0.165 | **0.148** | 0.372 | 0.381 | **0.370** |
| | MAE | 0.217 | 0.207 | **0.187** | 0.265 | 0.275 | **0.251** | 0.383 | 0.389 | **0.382** |
| 7 | MSE | 0.115 | 0.115 | **0.093** | 0.173 | 0.181 | **0.163** | 0.425 | 0.434 | **0.418** |
| | MAE | 0.243 | 0.236 | **0.211** | 0.280 | 0.290 | **0.266** | 0.415 | 0.420 | **0.411** |
| 30 | MSE | 0.143 | 0.142 | **0.115** | 0.195 | 0.207 | **0.187** | 0.452 | 0.447 | **0.443** |
| | MAE | 0.284 | 0.277 | **0.249** | 0.301 | 0.311 | **0.291** | 0.453 | 0.451 | **0.448** |
| 90 | MSE | 0.165 | 0.171 | **0.144** | 0.229 | 0.234 | **0.220** | 0.735 | **0.687** | 0.702 |
| | MAE | 0.311 | 0.314 | **0.291** | 0.332 | 0.337 | **0.322** | 0.622 | **0.600** | 0.600 |
| 180 | MSE | 0.189 | 0.204 | **0.176** | 0.251 | 0.257 | **0.240** | - | - | - |
| | MAE | 0.333 | 0.344 | **0.320** | 0.349 | 0.353 | **0.339** | - | - | - |

As shown in Table 8, for pre-training DERT, the separate mean prediction or Auto-correlation prediction task is effective. But combining them can obtain better date-representations. The mean prediction will lose some intraday information of time series, so we design the Auto-correlation prediction to mitigate the problem, it can induce DERT to tap some time series intraday information.

## F Training Details

There are 3 relatively independent components in Dateformer: DERT encoder, *global prediction*, and *local prediction*. They all can be trained separately. We found that Dateformers with different training stages strategies have different predictive performances. For some datasets, pre-training DERT or *global prediction* component is necessary. For other datasets, end-to-end training the entire Dateformer is also feasible. Thus, we design a 3-stage training methodology to tap the full capacity of Dateformer, and do the best on various forecast horizon demands of different datasets.

**Pre-training**   In section 2, we introduce 2 tasks to pre-train DERT, and the first step is to pre-train the DERT encoder. Time series are split into patches by day to supervise the learning of DERT. Each task contributes half of the pre-training loss. The pre-training stage aims to provide superior time representations as a container to distill and store global information from training set.

**Warm-up**   The second step is using the separately *global prediction* component to distill time series global information from training set, we call it warm-up phase. In warm-up, the pre-trained DERT encoder is loaded, then we train the separately *global prediction* component on training set. We insert this stage to force the *global prediction* component to remember the global characteristics of time series, so as to help Dateformer produce more robust long-range predictions. Furthermore, it also serves as the adapter when transferring a DERT from other datasets. This phase is optional, and we observed that a better short-term forecast is usually provided by the Dateformer skipping warm-up. More credible long-term predictions, however, always draw from the preheated one.

**Formal training** Dateformer loads pre-trained DERT or *global prediction* component then start training. We would use a small learning rate to fine-tune the pre-trained parameters. With enough memory, Dateformer can extrapolate to any number of days in the future, once trained.

Table 9: Multivariate time series forecasting comparisons of different training strategies, where Dateformer[11] goes through all 3 stages. Dateformer[10] skips the warm-up, and Dateformer[01] skips the pre-training stage. Dateformer[00] is end-to-end trained. The $*$ marked group of result is selected to compare with baseline models in the main text. The best results are highlighted in **bold**.

| Models | Dateformer[11] | | Dateformer[10] | | Dateformer[01] | | Dateformer[00] | |
|---|---|---|---|---|---|---|---|---|
| Metric | MSE | MAE | MSE | MAE | MSE | MAE | MSE | MAE |
| **PL(96)** | | | | | | | | |
| 1 | 0.042* | 0.141* | 0.032 | 0.119 | 0.044 | 0.142 | **0.027** | **0.112** |
| 3 | **0.076*** | **0.187*** | 0.107 | 0.220 | 0.078 | 0.192 | 0.077 | **0.176** |
| 7 | **0.093*** | **0.211*** | 0.147 | 0.271 | 0.097 | 0.219 | 0.122 | 0.227 |
| 30 | **0.115*** | **0.249*** | 0.275 | 0.375 | 0.133 | 0.269 | 0.324 | 0.384 |
| 90 | **0.144*** | **0.291*** | 0.600 | 0.587 | 0.166 | 0.311 | 0.798 | 0.679 |
| 180 | **0.176*** | **0.320*** | 0.911 | 0.733 | 0.218 | 0.358 | 1.238 | 0.843 |
| **ETTm1(96)** | | | | | | | | |
| 1 | 0.344 | 0.368 | **0.322*** | **0.355*** | 0.345 | 0.363 | 0.325 | 0.356 |
| 3 | 0.383 | 0.396 | **0.368*** | **0.381*** | 0.389 | 0.390 | 0.369 | 0.382 |
| 7 | 0.441 | 0.431 | **0.417*** | **0.410*** | 0.439 | 0.420 | 0.417 | 0.411 |
| 30 | 0.494 | 0.487 | **0.438*** | **0.446*** | 0.477 | 0.472 | 0.440 | 0.447 |
| 90 | 0.789 | 0.646 | **0.690*** | 0.600* | 0.723 | 0.614 | 0.693 | **0.598** |
| **ETTh2(24)** | | | | | | | | |
| 1 | 0.234 | 0.303 | **0.226** | **0.292** | 0.234* | 0.306* | 0.228 | 0.295 |
| 3 | 0.315 | 0.362 | **0.296** | **0.340** | 0.311* | 0.363* | 0.299 | 0.345 |
| 7 | 0.406 | 0.422 | **0.369** | **0.389** | 0.383* | 0.413* | 0.372 | 0.394 |
| 30 | 0.458 | 0.471 | 0.407 | 0.438 | 0.437* | 0.472* | **0.404** | **0.436** |
| 90 | 0.537 | 0.521 | 0.590 | 0.553 | **0.431*** | **0.486*** | 0.582 | 0.542 |
| **ECL(24)** | | | | | | | | |
| 1 | 0.113* | 0.218* | 0.101 | 0.198 | 0.121 | 0.231 | **0.100** | **0.198** |
| 3 | 0.148* | 0.251* | **0.146** | **0.237** | 0.158 | 0.265 | 0.156 | 0.241 |
| 7 | **0.163*** | 0.266* | 0.166 | **0.257** | 0.175 | 0.284 | 0.175 | 0.261 |
| 30 | **0.187*** | **0.291*** | 0.219 | 0.306 | 0.202 | 0.311 | 0.226 | 0.309 |
| 90 | **0.220*** | **0.322*** | 0.334 | 0.391 | 0.239 | 0.346 | 0.354 | 0.399 |
| 180 | **0.240*** | **0.339*** | 0.338 | 0.392 | 0.255 | 0.362 | 0.353 | 0.398 |
| **Traffic(24)** | | | | | | | | |
| 1 | 0.341 | 0.248 | 0.340 | 0.240 | 0.343* | 0.252* | **0.339** | **0.239** |
| 3 | 0.459 | 0.293 | 0.557 | 0.321 | **0.430*** | **0.284*** | 0.484 | 0.293 |
| 7 | 0.468 | 0.301 | 0.578 | 0.337 | **0.434*** | **0.289*** | 0.494 | 0.306 |
| 30 | 0.503 | 0.319 | 0.621 | 0.362 | **0.469*** | **0.308*** | 0.543 | 0.332 |
| 90 | 0.559 | 0.344 | 0.685 | 0.389 | **0.526*** | **0.339*** | 0.609 | 0.358 |
| **Weather(144)** | | | | | | | | |
| 1 | 0.224 | 0.294 | 0.223 | 0.295 | **0.220*** | 0.288* | 0.223 | **0.286** |
| 3 | 0.289 | 0.338 | 0.285 | 0.338 | **0.281*** | **0.329*** | 0.285 | 0.329 |
| 7 | 0.332 | 0.370 | 0.326 | 0.369 | **0.320*** | **0.360*** | 0.329 | 0.364 |
| 30 | 0.427 | 0.435 | 0.419 | 0.438 | 0.414* | 0.428* | **0.413** | **0.422** |
| 60 | 0.545 | 0.540 | 0.537 | 0.536 | 0.539* | 0.531* | **0.524** | **0.524** |
| **ER(1)** | | | | | | | | |
| 96 | 0.045 | 0.160 | **0.022*** | **0.107*** | 0.036 | 0.135 | 0.022 | 0.107 |
| 192 | 0.076 | 0.206 | **0.043*** | **0.152*** | 0.065 | 0.181 | 0.043 | 0.152 |
| 336 | 0.137 | 0.274 | **0.070*** | **0.195*** | 0.093 | 0.221 | 0.070 | 0.195 |
| 720 | 0.336 | 0.426 | **0.112*** | **0.255*** | 0.138 | 0.282 | 0.113 | 0.256 |

We use different training stages to train Dateformer, and results are shown in Table 9. It can be seen that different training stage setups can lead to different performances of Dateformers. There is no versatile training strategy does best on all forecast horizons of all datasets. Generally, the pre-training and warm-up can induce Dateformers to be more concerned with the global pattern of time series, and produce robust long-term predictions. For some datasets, before distilling global information from training set, pre-training a DERT encoder can enhance the distilling effect. But for other datasets, the 2 stages can be combined into one. Actually, the *global prediction* is also a fairly good pre-training task for DERT. The end-to-end trained Dateformer always contributes the best short-term predictions. Because we train Dateformer in the short-term forecasting task of 7d-predict-1d, the direct end-to-end training makes Dateformer more care local pattern of time series.

# G  PATCH SIZE CHOICE

In the main text, we choose the "day" as patch size, because it's moderate, close to people's habits, and convenient for modeling. Here, we try some other patch sizes. We select $\frac{1}{3}$ day, half-day, and 3-day as comparative patch sizes. The results are shown below.

Table 10: Multivariate forecast comparison of Dateformer using different patch sizes. "-" denotes that the patch size is too coarse to align with the forecast horizon.

| Time Series Datasets | | *ETTh2* | | | | *Traffic* | | | |
|---|---|---|---|---|---|---|---|---|---|
| Forecast Days | Metric | $\frac{1}{3}$ day | half-day | day | 3-day | $\frac{1}{3}$ day | half-day | day | 3-day |
| 1 | MSE | **0.224** | 0.226 | 0.234 | - | 0.421 | 0.417 | **0.343** | - |
|   | MAE | 0.315 | 0.315 | **0.306** | - | 0.285 | 0.283 | **0.252** | - |
| 3 | MSE | 0.325 | 0.318 | **0.311** | 0.366 | 0.558 | 0.526 | **0.430** | 0.472 |
|   | MAE | 0.382 | 0.374 | **0.363** | 0.422 | 0.379 | 0.326 | **0.284** | 0.311 |
| 7 | MSE | 0.428 | 0.407 | **0.383** | - | 0.564 | 0.524 | **0.434** | - |
|   | MAE | 0.452 | 0.429 | **0.413** | - | 0.384 | 0.328 | **0.289** | - |
| 30 | MSE | 0.513 | 0.458 | **0.437** | 0.690 | 0.627 | 0.548 | **0.469** | 0.527 |
|    | MAE | 0.541 | 0.483 | **0.472** | 0.722 | 0.455 | 0.340 | **0.308** | 0.330 |
| 90 | MSE | 0.540 | 0.485 | **0.431** | 0.676 | 0.655 | 0.582 | **0.526** | 0.564 |
|    | MAE | 0.577 | 0.505 | **0.486** | 0.715 | 0.447 | 0.360 | **0.339** | 0.361 |

As shown in Table 10, for some time series, finer patch sizes can lead to better short-term predictions, because the closer time series carries more accurate local pattern information. Compared to a day ago, the local pattern of the present is more similar to it of half a day ago. But their mid-long-term predictions deteriorate, because finer patch sizes will result in more *local predictions* recursions for the same size forecast horizons, which means more error accumulation. In addition, for these time series whose daily periodicity is dominant (like *Traffic*), the "day" is the best patch size. These time series are usually closely related to human activities and hence more concerned with daily patterns. The coarse patch size beyond "day" is not the most important activity period of people, so it's difficult to construct sufficiently accurate time-representations. If there is a holiday in the 3-day patch, how to embed which day it is in the time-embedding by the method mentioned at the beginning of Section 2? That leads over-coarse patch sizes inapplicable. For most time series related to human activity, we recommend the "day" as the patch size, it's moderate, close to people's habits.

# H  HYPER PARAMETER SENSITIVITY

We check the robustness with respect to the hyper-parameters: pre-days and post-days paddings. We select 6 groups of paddings: (1, 1), (3, 3), (7, 7), (7, 14), (14, 14) and (30, 30). To be more intuitive, we test the *global prediction* component with above several paddings on *PL* dataset.

Table 11: Dateformer's multivariate *global prediction* results on *PL* dataset using several paddings.

| Paddings | 1, 1 | | 3, 3 | | 7, 7 | | 7, 14 | | 14, 14 | | 30, 30 | |
|---|---|---|---|---|---|---|---|---|---|---|---|---|
| Metric | MSE | MAE | MSE | MAE | MSE | MAE | MSE | MAE | MSE | MAE | MSE | MAE |
| 1 | 0.155 | 0.290 | 0.151 | 0.285 | **0.127** | **0.264** | 0.134 | 0.270 | 0.139 | 0.278 | 0.156 | 0.291 |
| 3 | 0.155 | 0.290 | 0.152 | 0.286 | **0.128** | **0.264** | 0.134 | 0.271 | 0.139 | 0.278 | 0.157 | 0.292 |
| 7 | 0.156 | 0.291 | 0.153 | 0.287 | **0.128** | **0.265** | 0.134 | 0.271 | 0.139 | 0.278 | 0.157 | 0.293 |
| 30 | 0.161 | 0.296 | 0.158 | 0.292 | **0.132** | **0.269** | 0.137 | 0.274 | 0.142 | 0.281 | 0.161 | 0.297 |
| 90 | 0.179 | 0.315 | 0.176 | 0.311 | **0.145** | **0.284** | 0.151 | 0.291 | 0.156 | 0.297 | 0.177 | 0.314 |
| 180 | 0.190 | 0.326 | 0.187 | 0.324 | **0.155** | **0.295** | 0.165 | 0.305 | 0.173 | 0.314 | 0.197 | 0.335 |

As shown in Table 11, too large or too small padding sizes will make the prediction performance worse. The moderate padding size (7, 7) leads to the best *global prediction*. This is also consistent with how people generally think—we don't simply consider the here and now, but also don't think

too far ahead. In the main text, to balance predictive performances between various time series datasets, we use the padding size of (7, 14) instead of the best (7, 7).

## I   IMPLEMENTATION DETAILS

Our proposed models are trained with L2 loss, and using AdamW (Loshchilov & Hutter, 2017) optimizer with weight decay of $7e^{-4}$. We adjust learning rate by OneCycleLR (Smith & Topin, 2019) which use 3 phase scheduling with the percentage of the cycle spent increasing the learning rate is 0.4, and the max learning rate is $1e^{-3}$. Batch size for pre-training is 64, and 32 for others. The total epochs are 100, but the models normally super converges very quickly. All experiments are repeated 3 times, implemented by PyTorch(Paszke et al., 2019), and trained on a NVIDIA RTX3090 24GB GPUs. Numbers of the pre-days and post-days padding are 7 and 14 respectively. There is 1 layer in DERT encoder, and the inside Transformer contains 4 layers both in encoder and decoder.

The implementations of Autoformer (Wu et al., 2021), Informer (Zhou et al., 2021), and Reformer (Kitaev et al., 2020) are from the Autoformer's repository [11]. And the implementations of FED-former[12] (Zhou et al., 2022) and Pyraformer[13] (Liu et al., 2021) are from their respective repository. We adopt the hyper-parameters setting that recommended in their repositories but unify the token's dimension $d_{model}$ as 512. We fix the input series length as 96 time-steps for FEDformer, Autoformer, Informer, and Reformer. This is recommended by them or empirical results, and we found extending their input length will result in unstable performances of these baselines (see section E.1). For Pyraformer, facing longer forecast horizons, we extend its input length as their paper recommended. For more detailed hyper-parameters setting please refer to their code repositories. For other baseline models, we also use grid-search method to select their hyper-parameters.

## J   COMPLEXITY ANALYSIS

We also provide the complexity analysis of the *global prediction* and *local prediction* components. For an input lookback window with length $L_i$ and output forecast window with length $L_o$, they're all divisible by $G$ in the setting of splitting time series into patches by day, where $G$ denotes time-steps number every day of the time series datasets. The complexity of *global prediction* component is $\mathcal{O}(L_i + L_o)$. For *local prediction* component, it recurses $\frac{L_o}{G}$ times and the time complexity of each time shall not exceed $\mathcal{O}((\frac{L_i+L_o}{G})^2)$. So, the time complexity of *local prediction* component is $\mathcal{O}(\frac{L_o}{G}(\frac{L_i+L_o}{G})^2)$ at most. Its maximum memory usage is $\mathcal{O}(L_i + L_o + (\frac{L_i+L_o}{G})^2)$, where $(\frac{L_i+L_o}{G})^2$ happens in the last day's prediction. In practice, $G^2$ will be a big number for fine-grained time series and the $L_i + L_o$ takes up the most memory, it becomes the main factor that restricts us to predict further away. Under 24GB memory, Dateformer's maximum forecast horizon exceeds all baseline models. And its inference speed is just slightly slower than Transformers that adopts one-step forward generative style inference (Zhou et al., 2021) because the *local prediction* component requires a few recursions.

---

[11]https://github.com/thuml/Autoformer

[12]https://github.com/MAZiqing/FEDformer

[13]https://github.com/alipay/Pyraformer

# K MAIN RESULTS WITH STANDARD DEVIATIONS

We repeat all experiments 3 times, and the results with standard deviations are shown in Table 12.

Table 12: Quantitative results with fluctuations of different forecast days for multivariate forecast.

| Models | | Dateformer | | FEDformer | | Autoformer | | Informer | | Pyraformer | |
|---|---|---|---|---|---|---|---|---|---|---|---|
| Metric | | MSE | MAE | MSE | MAE | MSE | MAE | MSE | MAE | MSE | MAE |
| PL(96) | 1 | **0.042**±0.003 | 0.141±0.006 | 0.105±0.001 | 0.231±0.003 | 0.098±0.013 | 0.201±0.017 | 0.067±0.002 | 0.157±0.003 | 0.052±0.002 | **0.139**±0.002 |
| | 3 | **0.076**±0.001 | **0.187**±0.001 | 0.149±0.004 | 0.264±0.005 | 0.405±0.022 | 0.479±0.018 | 0.251±0.045 | 0.331±0.015 | 0.134±0.003 | 0.230±0.002 |
| | 7 | **0.093**±0.001 | **0.211**±0.002 | 0.196±0.009 | 0.303±0.007 | 0.398±0.096 | 0.462±0.073 | 0.370±0.014 | 0.424±0.005 | 0.211±0.005 | 0.313±0.001 |
| | 30 | **0.115**±0.004 | **0.249**±0.005 | 0.423±0.020 | 0.471±0.010 | 0.902±0.371 | 0.727±0.182 | 0.373±0.013 | 0.458±0.013 | 0.268±0.003 | 0.381±0.003 |
| | 90 | **0.144**±0.003 | **0.291**±0.004 | OOM | OOM | OOM | OOM | OOM | OOM | 0.585±0.001 | 0.590±0.004 |
| | 180 | **0.176**±0.004 | **0.320**±0.004 | OOM | OOM | OOM | OOM | OOM | OOM | OOM | OOM |
| ETTm1(96) | 1 | **0.322**±0.008 | **0.355**±0.004 | 0.341±0.005 | 0.390±0.004 | 0.491±0.046 | 0.476±0.014 | 0.640±0.058 | 0.570±0.024 | 0.515±0.036 | 0.505±0.020 |
| | 3 | **0.368**±0.006 | **0.381**±0.004 | 0.419±0.008 | 0.439±0.001 | 0.598±0.029 | 0.522±0.014 | 0.963±0.082 | 0.750±0.042 | 0.849±0.061 | 0.713±0.027 |
| | 7 | **0.417**±0.006 | **0.410**±0.003 | 0.473±0.003 | 0.468±0.004 | 0.646±0.075 | 0.547±0.026 | 1.129±0.051 | 0.800±0.040 | 1.053±0.010 | 0.818±0.007 |
| | 30 | **0.438**±0.002 | **0.446**±0.001 | 0.547±0.029 | 0.525±0.018 | 0.681±0.031 | 0.581±0.009 | 1.132±0.005 | 0.831±0.004 | 1.020±0.013 | 0.806±0.009 |
| | 90 | **0.690**±0.019 | **0.600**±0.007 | OOM | OOM | OOM | OOM | OOM | OOM | 1.056±0.018 | 0.845±0.009 |
| ETTh2(24) | 1 | **0.234**±0.001 | **0.306**±0.002 | 0.246±0.012 | 0.327±0.010 | 0.289±0.010 | 0.364±0.007 | 1.606±0.198 | 0.997±0.060 | 0.412±0.030 | 0.498±0.018 |
| | 3 | **0.311**±0.005 | **0.363**±0.002 | 0.334±0.011 | 0.381±0.003 | 0.347±0.008 | 0.395±0.004 | 1.928±0.283 | 1.111±0.087 | 1.140±0.033 | 0.832±0.014 |
| | 7 | **0.383**±0.010 | **0.413**±0.005 | 0.412±0.004 | 0.426±0.002 | 0.451±0.019 | 0.451±0.014 | 6.200±0.470 | 2.024±0.065 | 4.877±0.752 | 1.747±0.158 |
| | 30 | **0.437**±0.015 | **0.472**±0.002 | 0.466±0.010 | 0.483±0.003 | 0.510±0.004 | 0.511±0.004 | 4.091±0.180 | 1.717±0.032 | 4.674±0.293 | 1.869±0.055 |
| | 90 | **0.431**±0.033 | **0.486**±0.008 | 0.719±0.035 | 0.618±0.017 | 0.702±0.056 | 0.631±0.032 | 2.571±0.023 | 1.217±0.011 | 3.330±0.036 | 1.511±0.014 |
| ECL(24) | 1 | **0.113**±0.002 | **0.218**±0.003 | 0.169±0.001 | 0.288±0.001 | 0.174±0.006 | 0.293±0.007 | 0.328±0.009 | 0.412±0.006 | 0.268±0.006 | 0.372±0.006 |
| | 3 | **0.148**±0.002 | **0.251**±0.003 | 0.186±0.001 | 0.302±0.001 | 0.215±0.024 | 0.329±0.020 | 0.358±0.006 | 0.430±0.006 | 0.308±0.007 | 0.388±0.004 |
| | 7 | **0.163**±0.002 | **0.266**±0.003 | 0.201±0.002 | 0.316±0.002 | 0.208±0.007 | 0.320±0.006 | 0.387±0.009 | 0.459±0.006 | 0.281±0.008 | 0.381±0.008 |
| | 30 | **0.187**±0.001 | **0.291**±0.002 | 0.242±0.002 | 0.351±0.003 | 0.259±0.005 | 0.364±0.005 | 0.400±0.008 | 0.460±0.004 | 0.288±0.005 | 0.382±0.004 |
| | 90 | **0.220**±0.003 | **0.322**±0.003 | 0.316±0.008 | 0.404±0.006 | 0.382±0.066 | 0.439±0.033 | 0.550±0.019 | 0.552±0.011 | OOM | OOM |
| | 180 | **0.240**±0.002 | **0.339**±0.003 | - | - | - | - | - | - | - | - |
| Traffic(24) | 1 | **0.343**±0.004 | **0.252**±0.001 | 0.547±0.004 | 0.357±0.004 | 0.554±0.006 | 0.363±0.008 | 0.678±0.018 | 0.383±0.017 | 0.600±0.002 | 0.337±0.001 |
| | 3 | **0.430**±0.007 | **0.284**±0.003 | 0.581±0.003 | 0.367±0.002 | 0.625±0.041 | 0.391±0.025 | 0.721±0.023 | 0.404±0.013 | 0.635±0.005 | 0.358±0.005 |
| | 7 | **0.434**±0.006 | **0.289**±0.002 | 0.613±0.001 | 0.382±0.001 | 0.705±0.009 | 0.439±0.008 | 0.768±0.021 | 0.425±0.011 | 0.639±0.005 | 0.356±0.004 |
| | 30 | **0.469**±0.004 | **0.308**±0.002 | 0.652±0.001 | 0.395±0.003 | 0.686±0.020 | 0.420±0.011 | 0.959±0.030 | 0.534±0.017 | OOM | OOM |
| | 90 | **0.526**±0.002 | **0.339**±0.001 | - | - | - | - | - | - | - | - |
| Weather(144) | 1 | **0.220**±0.001 | **0.288**±0.002 | 0.234±0.001 | 0.304±0.002 | 0.327±0.023 | 0.377±0.017 | 0.370±0.029 | 0.424±0.022 | 0.231±0.011 | 0.310±0.011 |
| | 3 | **0.281**±0.002 | **0.329**±0.002 | 0.338±0.001 | 0.375±0.004 | 0.370±0.009 | 0.402±0.006 | 0.628±0.008 | 0.560±0.006 | 0.388±0.011 | 0.427±0.010 |
| | 7 | **0.320**±0.005 | **0.360**±0.001 | 0.495±0.112 | 0.472±0.068 | 0.474±0.044 | 0.461±0.035 | 1.093±0.057 | 0.768±0.024 | 0.442±0.007 | 0.460±0.002 |
| | 30 | **0.414**±0.005 | **0.428**±0.003 | 0.688±0.021 | 0.580±0.011 | 0.724±0.008 | 0.604±0.005 | 2.789±0.326 | 1.303±0.061 | 0.823±0.007 | 0.676±0.005 |
| | 60 | **0.539**±0.010 | **0.531**±0.003 | - | - | - | - | - | - | - | - |
| ER(1) | 96 | **0.022**±0.001 | **0.107**±0.001 | 0.041±0.002 | 0.154±0.004 | 0.040±0.001 | 0.154±0.003 | 0.248±0.005 | 0.361±0.003 | 0.194±0.006 | 0.340±0.005 |
| | 192 | **0.043**±0.001 | **0.152**±0.001 | 0.062±0.001 | 0.194±0.002 | 0.061±0.002 | 0.193±0.004 | 0.430±0.063 | 0.474±0.030 | 0.337±0.013 | 0.440±0.008 |
| | 336 | **0.070**±0.001 | **0.195**±0.001 | 0.087±0.002 | 0.233±0.005 | 0.096±0.007 | 0.246±0.008 | 0.756±0.056 | 0.642±0.028 | 0.607±0.110 | 0.600±0.060 |
| | 720 | **0.112**±0.001 | **0.255**±0.001 | 0.165±0.006 | 0.317±0.005 | 0.389±0.313 | 0.422±0.131 | 1.073±0.021 | 0.773±0.011 | 0.963±0.030 | 0.772±0.011 |

# L SHOWCASES

In order to more intuitively show Dateformer's prediction results, we visualized the time series ground-truth and predictions of several forecast tasks. The charts are shown as follows.

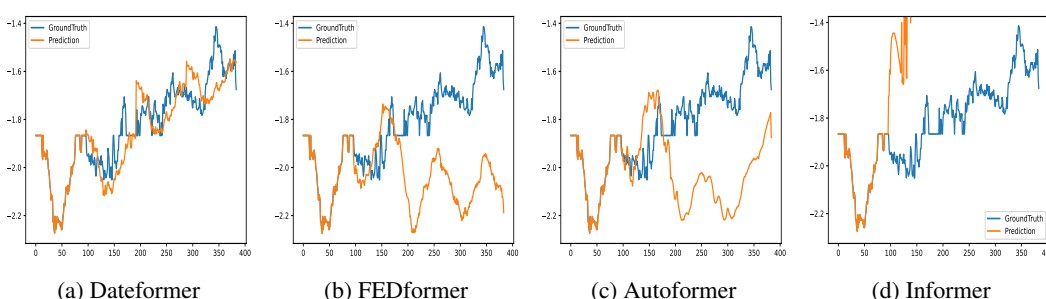

| (a) Dateformer | (b) FEDformer | (c) Autoformer | (d) Informer |
|---|---|---|---|

Figure 8: 3 days prediction cases from *ETTm1* oil temperature series

It can be seen that Dateformer's predictions are closest to the ground-truth. Compared to other models, Dateformer accurately grasps time series' global pattern: e.g., overall trend and long-range seasonality, that is what other models are not good at. Because they can only analyze lookback window series to predict. But the global pattern should be captured from whole training set series.

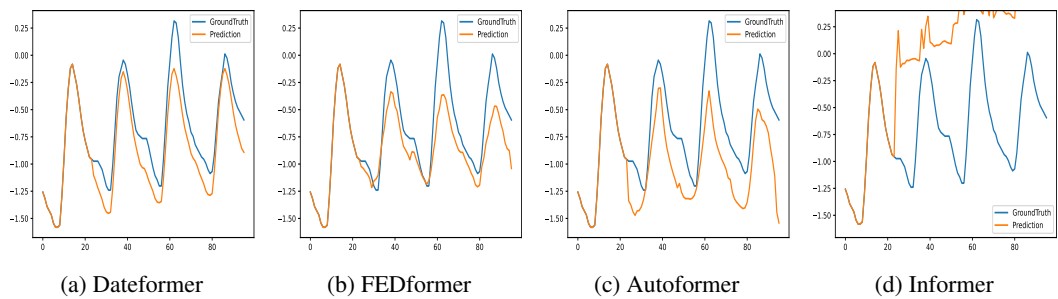

Figure 9: 3 days prediction cases from *ETTh2* oil temperature series

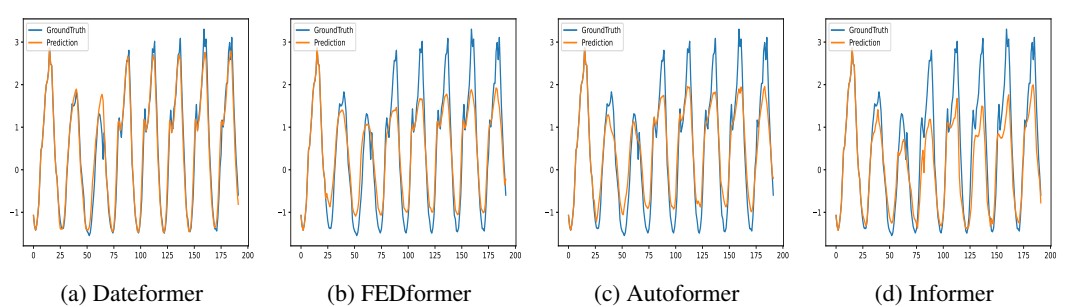

Figure 10: 7 days prediction cases from *Traffic* series

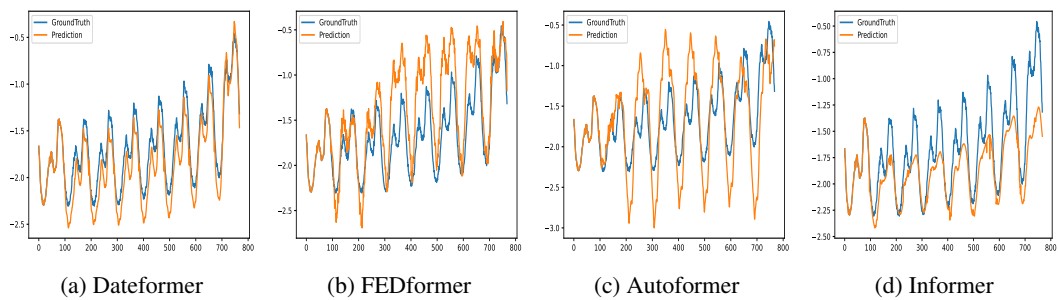

Figure 11: 7 days prediction cases from *P*ower *L*oad series

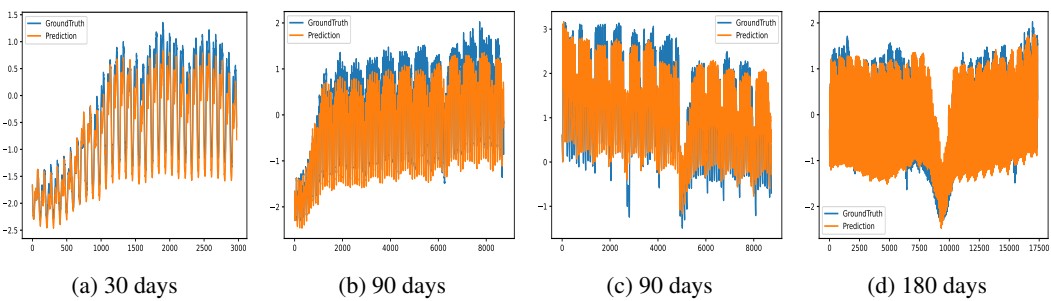

Figure 12: Mid-long-term prediction cases of Dateformer from *P*ower *L*oad series

# M LOCAL PREDICTIONS WITH GENERATIVE STYLE

In Section 3.2, we introduce Dateformer's *local prediction* component that implemented by the vanilla Transformer. But note that the vanilla Transformer is just to produce similarities to aggregate series residuals, it does not directly generate *local prediction*. If we employ a *local prediction* component with generative style, how does Dateformer perform? As a basic comparison, we modify Dateformer's *local prediction* component to be generative style.

- **Dateformer-GD**: removing Equation 4's FFN and $\mathrm{SoftMax}$ that calculate similarities and attaching a FFN to $\widehat{\boldsymbol{d}_{P+1}}$ to directly generate the *local prediction* corresponding $\boldsymbol{d}_{P+1}$.
- **Dateformer-GR**: inputting $Date\ Representations$ to the Transformer's encoder and inputting $Series\ Residuals$ to the Transformer's decoder. For the decoder's output $\{\widehat{\boldsymbol{r}_1}, \widehat{\boldsymbol{r}_2}, \cdots, \widehat{\boldsymbol{r}_P}\}$, we use:

$$\widehat{\boldsymbol{r}} = \mathrm{AveragePooling}(\widehat{\boldsymbol{r}_1}, \widehat{\boldsymbol{r}_2}, \cdots, \widehat{\boldsymbol{r}_P})$$
$$Local\ Prediction = \mathrm{FFN}(\widehat{\boldsymbol{r}}) \tag{8}$$

to directly generate the *local prediction* corresponding $\boldsymbol{d}_{P+1}$.

Table 13: Multivariate forecast comparison of several Dateformer variants. "-" denotes lacking test samples to report result.

| Time Series Datasets | | *PL* | | | *Traffic* | | |
|---|---|---|---|---|---|---|---|
| Forecast Days | Metric | Ours | GD | GR | Ours | GD | GR |
| 1 | MSE | **0.042** | 0.047 | 0.047 | **0.343** | 0.639 | 0.642 |
| | MAE | **0.141** | 0.153 | 0.149 | **0.252** | 0.384 | 0.387 |
| 3 | MSE | **0.076** | 0.093 | 0.087 | **0.430** | 0.653 | 0.653 |
| | MAE | **0.187** | 0.211 | 0.206 | **0.284** | 0.393 | 0.393 |
| 7 | MSE | **0.093** | 0.127 | 0.109 | **0.434** | 0.660 | 0.657 |
| | MAE | **0.211** | 0.252 | 0.237 | **0.289** | 0.396 | 0.394 |
| 30 | MSE | **0.115** | 0.218 | 0.144 | **0.469** | 0.675 | 0.675 |
| | MAE | **0.249** | 0.350 | 0.282 | **0.308** | 0.399 | 0.400 |
| 90 | MSE | **0.144** | 0.297 | 0.170 | **0.526** | 0.680 | 0.685 |
| | MAE | **0.291** | 0.433 | 0.314 | **0.339** | 0.404 | 0.409 |
| 180 | MSE | **0.176** | 0.464 | 0.177 | - | - | - |
| | MAE | **0.320** | 0.540 | 0.322 | - | - | - |

As shown in Table 13, the *local prediction* component with aggregating style always outperforms it with generative style. Due to splitting time series into patches, training samples considerably reduce. That makes the *local prediction* component with generative style easy to overfit. To mitigate the problem, we introduce an inductive bias: the local patterns of adjacent time series are similar and hence design the *local prediction* component with aggregated style to aggregate similar local pattern information.

# N  OTHER MODEL'S REMAINDERS

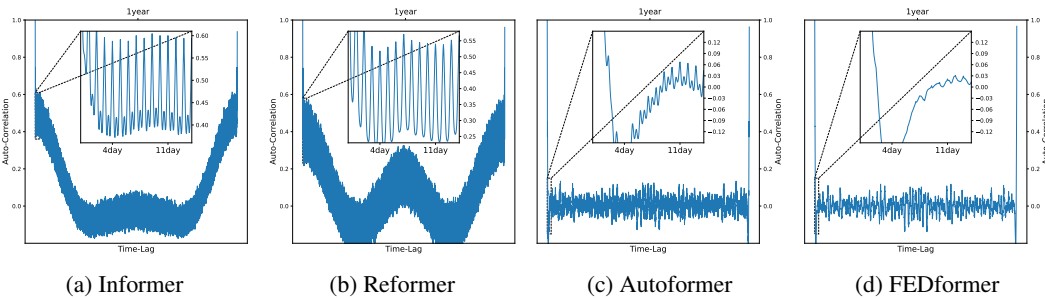

(a) Informer     (b) Reformer     (c) Autoformer     (d) FEDformer

Figure 13: Auto-correlation of 4 time series remainders from *ETT* oil temperature series.

We show 4 auto-correlation series in Figure 5, to provide a intuitive view to the characteristic of global and local pattern. That's not for comparison with other models, just to intuitively explain each component's function.

But as additional interests, we also provide auto-correlation series of other Transformers' remainders here. As shown in Figure 13a and 13b, the auto-correlation series of the remainders produced by Informer and Reformer did not drop to 0 immediately at the left end, which indicates they fail to accurately predict time series' mean. And the remainders of Informer, Reformer, and Autoformer still exhibit obvious seasonality that they didn't capture. Referring to Figure 13d, though not obvious, there is also weak seasonality in FEDformer's remainder.

