# OpenReview forum: "Dateformer: Transformer Extends Look-back Horizon to Predict Longer-term Time Series"
_ICLR.cc/2023/Conference — Submitted to ICLR 2023_

### Official Review · Reviewer_atRd · 2022-10-21

**Confidence:** 3
**Correctness:** 3
**Technical Novelty And Significance:** 3
**Empirical Novelty And Significance:** 3
**Recommendation:** 6

**Clarity, Quality, Novelty And Reproducibility:**

Clarity : This work is somewhat clear although there are some parts (such as those mentioned earlier) that could be made more clear


Quality : The work proposes a well structured approach. However, there are several questions that come up.


Novelty : This work is somewhat novel. It does use some known mechanisms but also discusses interesting ideas such as global predictions from date representations. The combined use in the overall framework is somewhat original.


Reproducibility : The manuscript claims that the code would be released soon.

**Strength And Weaknesses:**

Strength :

This work proposes a structured approach and uses several concepts to create a framework that can be utilized with a very long history to forecast time-series. Furthermore, the performance over several datasets demonstrates the strength of the proposed approach. Overall, this work does seem in the right direction for more robust time-series forecasting. However, there are some questions that come up.


Weakness :

There seem to be some ideas that were not very clear. Such as :

--- As a basic comparison, what would be the performance if the aggregate value (or a domain/data based learned representation) of each patch (a day in this case) is directly used as input to a transformer based forecasting model? The length of the input would be the same as what is proposed in this work so any additional resource issue would be mitigated.

--- Why is it that broadening the window will exhaust model capacity. A sampling approach can be used to pick samples from a broad window and these samples can then behave like a narrow window.


--- I am not sure I fully understand Figure 1. Both 1a and 1b seem to have similar trends. Therefore, I am not sure about the exact pitch of Figure 1.

--- Why should equivalent token information be maintained across time-series. It might be beneficial to have variable patch sizes based on the information. Such an approach is useful especially to take into account unexpected events (such as a storm that causes power failure).

--- What is the ground truth and loss used for training the global prediction model ? Is it a time-series value? Does this mean that a time-series value (e.g., power load) is being predicted using generic date representations. I think that I might have misunderstood and perhaps a clarification would be useful.

--- Most of the datasets are slow moving (except perhaps the economics data). It would be beneficial to gauge the performance of the proposed approach on fast moving data that still has global and local characteristics. This would perhaps again bring into question if using a day as the patch size is beneficial.

**Summary Of The Paper:**

This work introduces Dateformer for time-series forecasting. The time-series is split into day based patches and the processing is shifted from time-based towards patch based. This work uses time representations as the modeling entity and empirically demonstrates the strength of the proposed approach on multiple datasets.

**Summary Of The Review:**

Overall, this work seems to be in the correct direction for utilizing local and long term global information for time-series forecasting. A structured approach and empirical explorations are provided to strengthen the overall pitch. However, there are some questions that come up such as the use of a fixed size (day based) patch , variations in the type of data between the datasets, and some basic baselines.

---

> ### Author Response · Authors · 2022-11-13
> **Response to Reviewer atRd (Part1/2)**
>
> Thank you very much for your insightful comments and recognition about our study's overall direction. We can't agree more that long-term series forecasting should utilize local and long-term global information. We are very sorry for some unclear expressions in the paper. We have followed your helpful opinion to improve our paper to be clearer. The updated paper will be uploaded later.
>
> **Q1: As a basic comparison, what would be the performance if the aggregate value (or a domain/data based learned representation) of each patch (a day in this case) is directly used as input to a transformer based forecasting model?**
>
> That's a really good question! In fact, initially we input the aggregate value or date-representation to Transformer for directly generating local prediction rather than aggregating them. The results are shown below.
>
> *Multivariate forecast comparison on Traffic. Only MSE is reported.*
> |  Model Variant   | predict 1 day | predict 3 | predict 7 | predict 30 | predict 90 days |
> |  :-----:  | :-----:  | :-----: | :-----: | :-----: | :-----: |
> | Input Aggregate Value | 0.642 | 0.653 | 0.657 | 0.675 | 0.685 |
> | Input Date-Representation | 0.639 | 0.653 | 0.660 | 0.675 | 0.680 |
> | Aggregate(ours) | **0.343** | **0.430** | **0.434** | **0.469** | **0.526** |
>
> It can be seen, inputting aggregate value/date-representation to directly generating prediction is not as good as the aggregating value. We think it's because the local prediction component with generative style is very easy to overfit training set. Splitting time series into patches leads to training sample reduce a lot. So, we introduce an inductive bias that the local patterns of adjacent time series are similar and hence design the aggregating style local prediction component. We add the comparison and more details to Appendix M of the updated paper.
>
> **Q2: Why is it that broadening the window will exhaust model capacity. A sampling approach can be used to pick samples from a broad window and these samples can then behave like a narrow window.**
>
> We generally agree with you that sampling is a good approach to extend models' lookback horizon. But compared to splitting time series into patches, sampling will lose more information. For example, given a time series recorded per 15 min, splitting time series into patches by day will transform 96 time-steps to a token and preserving most information. But for sampling, if sample 1 time-step from 96 will lose most information of the day. The finer time series' granularity, the greater the information gap.
>
> On the other hand, at present, there are few Transformer-based forecasting research using sampling, so we didn't discuss it in our paper. Mainstream predictive Transformers will exhaust model capacity as broadening lookback/forecast windows. We believe that the combination of Transformer and sampling is a promising direction. But it's unnecessary to be discussed in our paper.
>
> **Q3: I am not sure I fully understand Figure 1. Both 1a and 1b seem to have similar trends. Therefore, I am not sure about the exact pitch of Figure 1.**
>
> We are very sorry that our unclear statement confused you.  We have added more detailed explanation to Figure1 in the updated paper.
>
> The Figure1a is to intuitively show time series' global pattern. Refer to Figure1a, models observe an obvious upward trend in the zoom-in window. But zoom out, we know that's a yearly seasonality. And we can see a slightly overall upward trend between the 2 years power load series. Reviewer Q1u4 also appreciate the chart and commend it for its informative.
>
> The Figure1b to intuitively show date's polysemy thereby introducing DERT and dynamic date-representation in Section 2. Refer to Figure1b: compared to a week ago or later, the power load series of Jan 2, 2012, is closer to a year ago, which indicates the day's time semantics is altered to closer to a year ago but further away from a week ago or later.
>
> **Q4: Why should equivalent token information be maintained across time-series. It might be beneficial to have variable patch sizes based on the information. Such an approach is useful especially to take into account unexpected events.**
>
> Because the patch size "day" is moderate, close to people's habits, and convenient for modeling. We can also choose other patch size and you can refer to **Q1** of Reviewer Q1u4 or Appendix G of the updated paper for the details. We very agree that finer patch sizes can help model capture unexpected events information. However, variable or finer patch sizes require much more modeling cost, and make the model much more complex, but trade limited improvement: just short-term forecasting benefits, which is not in line with the paper's main idea: long-term forecasting. The unexpected events have little impact on long-term forecasting. That paying too much attention to it is not worthwhile. The patch size "day" is very close to people's habits, and convenient for modeling. So, its cost-effective.

---

> > ### Author Response · Authors · 2022-11-13
> > **Response to Reviewer atRd (Part 2/2)**
> >
> > **Q5: What is the ground truth and loss used for training the global prediction model? Is it a time-series value? Does this mean that a time-series value (e.g., power load) is being predicted using generic date representations. I think that I might have misunderstood and perhaps a clarification would be useful.**
> >
> > Yes, your understanding is absolutely correct. We require the global prediction component only using date-representations to predict time series. Before formally training the whole Dateformer, we separately train the global prediction component to distill the whole training set's global pattern information. The whole training set's time series patches are used to train the global prediction component, each time series patch is a training sample. After the separate training, the general global pattern information will be distilled, but the volatile local pattern information won't. Because the time representation is general for the whole historical series.
> >
> > We are very sorry that our unclear statement confused you. And we have followed your suggestion adding more explanation about the distilling details in the updated paper.
> >
> > **Q6: Most of the datasets are slow moving (except perhaps the economics data). It would be beneficial to gauge the performance of the proposed approach on fast moving data that still has global and local characteristics. This would perhaps again bring into question if using a day as the patch size is beneficial.**
> >
> > Yes, the question is very insightful and professional. We must admit, the most important shortcoming of our model lies in its excessive dependence on date representation. That prevent applying the model on fast moving time series.  But we believe, our work provides a novel methodology to tackle long-term series forecasting. And in the future work, we can create appropriate representations for other patch sizes, thereby promoting our approach to fast moving time series.
> >
> > We are very grateful and admirable for your insightful, responsible, and expert review to our work. Your suggestions help us improve the paper a lot. Your question also inspired us to explore new research directions. If you have any additional questions, please be generous let's know, we would be happy to discuss further with you. If you feel our answer explains your questions well, could you please raise the score for our paper?

---

> ### Author Response · Authors · 2022-11-21
> **We are looking forward to your reply**
>
> Dear reviewer,
>
> Did our reply solve your questions? If so, could you please raise the score for our paper? If not, please feel free to let's know your concerns. We would be really happy to discuss further with you.
>
> Thanks,
>
> All authors

---

> ### Author Response · Authors · 2022-12-11
> **Final inquiry**
>
> Dear reviewer atRd,
>
> As there is only last 1 day left until the end of the author-reviewer discussion stage, we would like to ask again whether our paper revisions and responses have addressed your concerns and questions adequately.  If so, could you please raise the score for our paper? If not, please feel free to let's know your concerns. We would be happy to discuss further with you.
>
> best regard,
>
> All authors

---

### Official Review · Reviewer_Q1u4 · 2022-10-23

**Confidence:** 4
**Correctness:** 3
**Technical Novelty And Significance:** 3
**Empirical Novelty And Significance:** 3
**Recommendation:** 6

**Clarity, Quality, Novelty And Reproducibility:**

  This paper is written in OK quality and clarity. The figures and examples are very helpful in understanding the method proposed and the idea behind. However, some of the sections are not well organized, for example the `Related Work` section is in Appendix while experiments' implementation details are mentioned as a long paragraph in the main article. `DERT` seems to be a very important architecture in capturing data representation but was not mentioned in the introduction and it is a bit hard to connect to the later `Dateformer` architecture without checking back and forth. Other questions are proposed in the previous `Strength And Weaknesses` part.
	On the other hand, I think it's a relatively interesting and novel idea to gather global information from the training set and improve prediction accuracy in long term time series forecasting. Most of the previous methods learn the global trend only implicitly by training with short time series intercepted from the training set, and use date/time related information as covariates to construct point-wise inputs. Learning date representation globally can better capture long term patterns.



**Strength And Weaknesses:**

This paper proposed an interesting point of view that the current methods cannot well capture the global pattern in the training dataset, like the overall trend and long-term seasonality, etc.
Strengths:
1. Interesting idea of capturing the global trend of the dataset using time representation and pretrain over the training set.
2. The example given is very informative -- while local trends of the time series are very similar, the actual value of the time series is increasing each year. The figures are also well generated, very helpful in understanding the overall architecture.
3. Strong empirical results, showing improvements in various lengths of prediction, especially in long-term forecasting.
4. Interesting ablation study, demonstrating the date representation could even be transferable.

The paper could be better with the following questions resolved:
1. While `day` could be a very useful patch size, have you explored other patch sizes and how does the performance change with patch size?
2. It's interesting that global trend of time series can be captured via pretraining the `DERT` encoder and generate time representation for date. I'm wondering if the distance between the training dataset and the test dataset can impact the final accuracy. For example, if the training dataset includes data in 2013, will the testset sample in 2014 generally perform better than 2015?
3. How does the inference time / memory occupation of the method compare to previous baselines?
4. It appears not clear to me what other baselines' remainder of ground truth minus prediction looks like, compared to Figure 6(d). The claim of getting "white noise" seems not supported well for me. Can you please add figures for baselines?
5. This method seems to rely on date representation, in that case, does it work for different scales of time series, for example millisecond level / month or year level time series forcasting? Does changing to a different patch size help?

Some minor issues:
1. Section 1, paragraph 2, change `how do` to `how to`.
2. Page 2, first line, ditto, change `how do` to `how to`.
3. Page 3, first paragraph, this is talking about covariates, seems a bit unnecessary, can be mentioned in the appendix.
4. Section 3, paragraph 1, line 2, change `time series inherent characteristics` to `time series' inherent characteristics`
5. Figure 3 is taking a huge space in page 4 but it seems simple to describe or demonstrate in a smaller figure.
6. Figure 5 is a bit chaotic in describing how the autoregressive results are combined with global prediction -- the figure can be better organized to show the work flow.
7. Table 2, better explain what the bolded results mean.
8. Second 4, implementation details, change `There are 1 layer` to `There is 1 layer`.


**Summary Of The Paper:**

 This paper claimed that the global pattern in time series is not well captured in recent time series forecasting methods because they infer the future by analyzing the part of past sub-series closest to the present. To better leverage the global pattern and deal with long term time series forecasting, this paper proposed the following ideas:
  1. Split time series into patches to reduce token number and tackle long-term series forecasting.
  2. Distill time representation from training set, propose DERT, a Transformer based date encoder to generate the representation specific to a certain date in the dataset.
  3. Propose Dateformer, a time-modeling time series forecast framework based on Transformers, to combine the prediction from global time representations and local lookback window.

The paper's experiments show this method can achieve state-of-the-art results on multiple representative datasets. Especially for a long prediction window, this method can generate results far better than previous methods.

The contributions are two-fold:
1. Proposed to leverage a date representation to capture the global trend of the dataset, and combine with prediction from local look back window.
2. Created the `DERT` and `DateFormer` architecture to utilize the global information and achieved excellent empirical results, especially in long term forecasting.


**Summary Of The Review:**

Overall, this is a paper that proposed a date representation to better capture global trends in the training dataset. The idea is very interesting and empirical results show promising impact, especially in long term series forecasting. The paper is OK in writing and the idea is presented clearly. Ablation studies show that the global prediction is very critical in the final results and local residuals are helpful in short term prediction. Even though, there're a few incoherent parts in writing and some limitations due to the nature of date representation, this paper is meaningful in understanding and leveraging the global trend of time series. I recommend this paper to be accepted.

---

> ### Author Response · Authors · 2022-11-13
> **Response to Reviewer Q1u4 (Part1/2)**
>
> We sincerely thank you for carefully reading our paper and recognizing our novelty and contributions. From your expert comments, we believe you have a good understanding to our paper and your questions are sufficiently valuable. In addition, your suggestions about paper-organizing are very helpful, insightful, and thorough. We follow all your suggestions to improve our paper a lot, the updated paper will be uploaded later. And we sincerely hope for more suggestions from you, if you have. We would be happy to discuss more with you.
>
> **Q1: While ``day`` could be a very useful patch size, have you explored other patch sizes and how does the performance change with patch size?**
>
> We  try other several representative patch sizes: $\frac{1}{3}$day, half-day, and 3-day. The results are shown as below. And we add the discussion in Appendix G of the updated paper.
>
> *Multivariate forecast comparison on ETTh2. "-" denotes that the patch size is too coarse to align with the forecast horizon. Only MSE is reported.*
> |  patch size   | predict 1 day  | predict 3 days | predict 7 days | predict 30 days | predict 90 days |
> |  :----:  | :-----:  | :-----: | :-----: | :-----: | :-----: |
> | $\frac{1}{3}$day  | **0.224** | 0.325 | 0.428 | 0.513 | 0.540 |
> | half-day  | 0.226 | 0.318 | 0.407 | 0.458 | 0.485 |
> | day  | 0.234 | **0.311** | **0.383** | **0.437** | **0.431** |
> | 3-day  | - | 0.366 | - | 0.690 | 0.676 |
>
> *Multivariate forecast comparison on Traffic.*
> |  patch size   | predict 1 day  | predict 3 days | predict 7 days | predict 30 days | predict 90 days |
> |  :----:  | :-----:  | :-----: | :-----: | :-----: | :-----: |
> | $\frac{1}{3}$day  | 0.421 | 0.558 | 0.564 | 0.627 | 0.655 |
> | half-day  | 0.417 | 0.526 | 0.524 | 0.548 | 0.582 |
> | day  | **0.343** | **0.430** | **0.434** | **0.469** | **0.526** |
> | 3-day  | - | 0.472 | - | 0.527 | 0.564 |
>
> For ETTh2, finer patch sizes can lead to better short-term prediction, because the closer time series carries more accurate local pattern information. The local pattern of the present is more similar to it of half a day ago, compared to a day ago. But their mid-long-term predictions deteriorate, because finer patch sizes will result in more local predictions recursions for forecast horizons of the same size, which means more error accumulation.
>
> For these time series whose daily periodicity is dominant (like Traffic), the "day'' is the best patch size. These time series are usually closely related to human activities and hence more concerned with daily patterns.
>
> The coarse patch sizes beyond "day" are not the most important activity period of people, so it's difficult to construct sufficiently accurate time-representations for them. If there is a holiday in the 3-day patch, how to embed which day it is into the time-embedding by the method mentioned at the beginning of Section 2? That leads over-coarse patch sizes inapplicable.  So its performance obviously deteriorates. We recommend the "day" as the patch size,  because it's moderate, close to people's habits, and convenient for modeling.
>
> **Q2:  If the distance between the training dataset and the test dataset can impact the final accuracy. For example, if the training dataset includes data in 2013, will the test set sample in 2014 generally perform better than 2015?**
>
> Yes. To answer the question, we split PL's test set into 2 equal size parts: a part close to and a part away from training set.
>
> *Multivariate forecast comparison (final prediction). Only MSE is reported.*
> |  distance   | predict 1 day  | predict 3 days | predict 7 days | predict 30 days | predict 90 days | predict 180 days |
> |  :----:  | :-----:  | :-----: | :-----: | :-----: | :-----: | :-----: |
> | close  | **0.033** | **0.063** | **0.080** | **0.100** | **0.106** | **0.096** |
> | faraway  | 0.052 | 0.089 | 0.105 | 0.107 | 0.133 | 0.119 |
>
> *Multivariate forecast comparison (global prediction).*
> |  distance   | predict 1 day  | predict 3 days | predict 7 days | predict 30 days | predict 90 days | predict 180 days |
> |  :----:  | :-----:  | :-----: | :-----: | :-----: | :-----: | :-----: |
> | close  | **0.098** | **0.097** | **0.095** | **0.088** | **0.083** | **0.087** |
> | faraway  | 0.160 | 0.160 | 0.161 | 0.162 | 0.178 | 0.163 |
>
> It can be seen that predictions of the close part always better than the faraway part. We think that's because time series' local pattern is always changing, so as global pattern, though slower. The close future is more similar to the present and hence easier to predict, compared to the faraway future.  The changed global pattern may lead to inaccurate global prediction, so we designed local prediction component to aggregate recent time series residuals, thereby mitigating the problem. If the distance between training set's time and the present too long, we advise to re-train the model using the most current time series observations, better global prediction and final prediction can be expected.

---

> > ### Author Response · Authors · 2022-11-13
> > **Response to Reviewer Q1u4 (Part2/2)**
> >
> > **Q3: How does the inference time / memory occupation of the method compare to previous baselines?**
> >
> > For fine-grained time series, due to splitting time series into patches by day, the tokens of Dateformer are less than other Transformers. So, its memory usage is less than previous baselines. That's also proved by the comparison results of the multivariate prediction in our paper: when facing very long forecasting demands, other Transformer OOM, but our model still work. The finer the granularity of the time series, the greater the improvement of memory usage.  The inference time are slightly longer than Transformers that adopt one-step forward generative style, but much faster than other baselines. Overall, compared to previous SOTA Transformers, we sacrifice a little bit of inference time to trade the much smaller memory usage.
> >
> > But for coarse-grained time series (like ER), if using the same input lengths, the memory occupation will exceed previous Transformers. So we use narrower input windows for the ER dataset.  Thanks to the learned global pattern, Dateformer still outperforms other Transformers.
> >
> > We also provide complexity analyses about the proposed models, in Appendix J of the updated paper.
> >
> > Generally, for an input lookback window with length $L_i$ and output forecast window with length $L_o$, they're all divisible by $G$ in the setting of splitting time series into patches by day, where $G$ denotes time-steps number every day of the time series datasets. The complexity of global prediction component is $\mathcal{O}(L_i+L_o)$.
> > For local prediction component, it recurses $\frac{L_o}{G}$ times and the time complexity of each time shall not exceed $\mathcal{O}((\frac{L_i+L_o}{G})^2)$. So, the time complexity of local prediction component is $\mathcal{O}(\frac{L_o}{G}(\frac{L_i+L_o}{G})^2)$ at most. Its maximum memory usage is $\mathcal{O}(L_i+L_o+(\frac{L_i+L_o}{G})^2)$, where $(\frac{L_i+L_o}{G})^2$ happens in the last day's prediction. In practice, $G^2$ will be a big number for fine-grained time series and the $L_i+L_o$ takes up the most memory, it becomes the main factor limiting our prediction further.
> >
> > **Q4: It appears not clear to me what other baselines' remainder of ground truth minus prediction looks like, compared to Figure 6(d). The claim of getting "white noise" seems not supported well for me. Can you please add figures for baselines?**
> >
> > We add the figures for 4 SOTA baselines in the end of the updated paper. We must clarity that we use the Figures to provide a intuitive view to the characteristic of global and local pattern. **That's not for comparison with other models, just to intuitively explain each component's function.**
> >
> > **Q5:This method seems to rely on date representation, in that case, does it work for different scales of time series, for example millisecond level / month or year level time series forecasting? Does changing to a different patch size help?**
> >
> > For time series whose granularity finer than "day", our model will perform better and need not to changing patch size. In the paper, to obtain global prediction, we duplicate date-representation to $G$ copies and add sequential positional encoding to these copies, where $G$ is the dataset's time-steps number every day. The operation is to refine the date-representation to time-representations with finer time scale, thereby denoting each time-step in the day. For finer-granularity time series, we just need to adjust $G$ to the dataset's time-steps number every day. But for time series whose granularity coarser than "day", we can't find good enough time-representations for it, which may lead to some performance damage. We'll further study in future work.
> >
> > **Some typos and paper-organizing problem**
> >
> > We sincerely thank you again for carefully reading our paper and proposing suggestions. Your suggestions are really helpful. We follow all your suggestions, improving our paper to be clearer and more well-organized.
> >
> > * We remove the unnecessary first paragraph of page 3.
> > * We replace Figure3 with more concise formulas.
> > * We re-plot Figure5 in clearer workflow.
> > * We add explanation about the bolded result to Table 2
> > * We add detailed explanation about global/local pattern information and Figure1, and transfer implementation details to Appendix.
> > * We didn't detailed talk about `DERT` in INTRODUCTION because it's relatively independent supporting work. And the INTRODUCTION will be too lengthy if we discuss it in INTRODUCTION.
> >
> > Hope our reply can clarify your questions. And if you have extra questions, please feel free to let's know. We really hope to further discuss with you.

---

> ### Author Response · Authors · 2022-11-21
> **We are looking forward to your reply**
>
> Dear reviewer,
>
> Did our reply solve your questions? If so, could you please raise the score for our paper? If not, please feel free to let's know your concerns. We would be really happy to discuss further with you.
>
> Thanks,
>
> All authors

---

> ### Comment · Reviewer_Q1u4 · 2022-12-10
> **Thanks for the response**
>
> Dear authors:
>
> Thanks for the very detailed update! Pleased to see my questions answered and now I have a better picture of the design.
> The updated version clearly does better in clarity, I'm still supportive on recommending this paper to be accepted with more confidence.
>
> Reviewer Q1u4

---

> ### Author Response · Authors · 2022-12-10
> **Thanks for your very useful and responsible review**
>
> Dear reviewer Q1u4,
>
> We sincerely thank you for your thoughtful and helpful comments. We are very glad that our community has you as a patient and responsible reviewer.
>
> Thank you again for your recognition of our work. But if we understand correctly, the score 6 means ``weak accept'', which is not in line with your positive comments that **recommend to be accepted**. May I ask if there is any misunderstanding? Hum...  if not, please ignore it, that's not important.
>
> Communication with you is the happiest thing for me in this ICLR experience. I wish you all the best.
>
> Thanks,
>
> All authors

---

### Official Review · Reviewer_Xz1K · 2022-10-24

**Confidence:** 3
**Correctness:** 2
**Technical Novelty And Significance:** 2
**Empirical Novelty And Significance:** 2
**Recommendation:** 5

**Clarity, Quality, Novelty And Reproducibility:**

The paper is very poorly written, which make it difficult to understand the motivations, structure, training and value add of the proposed model. Couple of questions below for instance:

1. What is local/global information here? In which datasets can they be found?

2. What does patch-wise processing mean concretely? Can this be shown in an architectural diagram?

3. What does it mean to predict the daily mean? Is this the MSE of the average daily value of the observation, or simply the aggregate MSE per day?

4. Why the assumption that daily seasonal effects are dominant? One can construct multiple counter examples where daily effects do not apply (e.g. intraday traffic loads would depend on time of day). Does the model perform ONLY when seasonal effects dominate and are not applicable to general time series datasets?

5. What datasets are being used in the attention patterns for the ablation analysis of E.2.1? This is not stated at all.

6. Do the benchmark models get access to the same date/calendar inputs as Dateformer?


**Strength And Weaknesses:**

Strengths
---
Specialised components to encode seasonal relationships are relatively underexplored, make the idea behind fusing both temporal as well as seasonal relationships an interesting proposition.

Weaknesses
---

However, there are several key issues associated with the paper that make it difficult to recommend for acceptance in its current form:

1.	The paper is presented in a very unclear manner – with many ideas presented only in high-level terms and without explanation or citation, which makes detailed evaluation difficult. See section below for more.

2.	Lack of hyperparameter optimisation – all benchmarks and the dateformer are calibrated using default settings, which makes it difficult to determine whether underperformance is due to improper hyperparams. This is particularly the case for time series data, where optimal hyperparams can vary wildly between datasets?

3.	The two-step training process for Dateformer greatly increases the training time versus other benchmarks – can this be combined to be made end-to-end?


**Summary Of The Paper:**

The paper introduces the Dateformer – a new transformer-based architecture for time series forecasting which combines embeddings for calendar seasonality along with autoregressive inputs using specialised components for each.

**Summary Of The Review:**

Given the difficulties in evaluating the paper, I would suggest that the authors improve clarity of presentation prior to submission.

---

> ### Author Response · Authors · 2022-11-14
> **Response to Reviewer Xz1K (Part 1/3)**
>
> We kindly disagree with the reviewer’s assessment to our paper. In particular, the reviewer gave our paper the score “3” based on some misunderstands and groundless criticisms. We emphasize that **the reviewer's all criticisms are exaggerated even baseless**.  Below we objectively respond to all criticisms in detail.
>
> **Criticism1: The paper is presented in a very unclear manner – with many ideas presented only in high-level terms and without explanation or citation, which makes detailed evaluation difficult.**
>
> The criticism is exaggerated. We admit, there are some unclear expressions in our paper. But these statements are only a small part, which does not affect readers' understanding of the main idea of the paper. And as reflecting evidence: **all other 3 reviewers have accurately understood our paper and recognized our contributions.** Among them, reviewer Q1u4 praised the paper's informative example diagram, interesting idea, and ablation study. Reviewer atRd also accurately understood our work and proposed professional questions. Reviewer VVpa recognized our novelty and praised the paper's comprehensive experiments and SoTA performance. They recognized that our paper is written in OK quality and clarity and proposed helpful suggestions to improve our paper to be clearer. We disagree with the critical words "very unclear". It's exaggerated.
>
> On the other hand, in our paper, all ideas are presented in terms that are frequent in our time series field even the whole deep learning community.
>
> **Q1: What is local/global information here? In which datasets can they be found?**
>
> The local/global information represent the information about time series' local/global pattern. The global pattern includes but not limits to overall trend and long-term existing seasonality of time series.  We highlight its importance in the Paragraph 2 and Figure1a in INTRODUCTION. The local pattern indicates short-term change of time series, such as traffic congestion caused by road construction, power load surge caused by abnormal high temperature. We also talk about it at the beginning of Section 3: `The global pattern is derived from time series inherent characteristics and hence doesn’t fluctuate violently, while the local pattern is caused by some accidental factors such as sudden changes in the weather, so it’s not long-lasting. `
>
> The most intuitive is Figure1a in the paper: referring to Figure1a, models observe an obvious upward trend in the zoom-in window. But zoom out, we know that's a yearly seasonality. And we can see a slightly overall upward trend between the 2 years power load series. Reviewer Q1u4 also appreciate the example and commend it for its informative. You can find local/global pattern in numerous time series which include all datasets in our paper.
>
> The term "global/local" is frequent in the time series field, and they have been well accepted. A lot of studies are based on the term or concept, such as [1][2][3]. We also describe them in Paragraph 2 and Figure1a, the beginning of Section 3, Table 2, and Figure 6,.etc. And we have added more detailed explanations in the updated paper.
>
>
> **Q2: What does patch-wise processing mean concretely? Can this be shown in an architectural diagram?**
>
> The point-wise processing means each time-step value is transformed to a token of Transformer. Correspondingly, the patch-wise processing means every multiple time-steps values are transformed to a single token.  In our paper, we split time series into patches by day, in other words, each time series patch contains a day's time step value and is transformed to a single token. As stated in Paragraph 4 of INTRODUCTION, it can considerably reduce tokens and enhances information efficiency thereby enabling vanilla Transformer to predict long-term series. The navy-blue blocks in Figures 4 and 5 denote time series patches, and there are detailed describe in Section 3.1 and 3.2.
>
> The term "patch" is hot in the CV even the whole deep learning field, it's very very very simple, basic, and intuitive. So, we didn't explain it in verbose sentences or diagrams. The term is common sense or foundation for massive papers of deep learning, such as [4][5] in CV field, and [6] in time series field. We cited the related papers in Paragraph 5 of Related Works.
>
> # Reference
>
> [1] FEDformer: Frequency enhanced decomposed transformer for long-term series forecasting. ICML 2022
>
> [2] Enhancing the locality and breaking the memory bottleneck of transformer on time series forecasting. NIPS2019
>
> [3] DEPTS: Deep Expansion Learning for Periodic Time Series Forecasting[J]. ICLR 2022
>
> [4] Masked autoencoders are scalable vision learners. CVPR 2022
>
> [5] An image is worth 16x16 words: Transformers for image recognition at scale. ICLR 2021
>
> [6] Triformer: Triangular, Variable-Specific Attentions for Long Sequence Multivariate Time Series Forecasting--Full Version[J]. IJCAI 2022.

---

> > ### Author Response · Authors · 2022-11-14
> > **Response to Reviewer Xz1K (Part 2/3)**
> >
> > **Q3:What does it mean to predict the daily mean? Is this the MSE of the average daily value of the observation, or simply the aggregate MSE per day?**
> >
> > It means to predict the daily mean. The MSE is calculated using predicted daily mean and real observation. It's a pre-training task, which means it will be removed after pre-training and won't participate in the downstream forecasting task. So, we didn't aggregate MSE per day.   The pre-training is for producing dynamic date-representations. After pre-training, the dynamic date-representation will be used in the downstream forecasting task. The aggregating is to make local predictions using the produced dynamic date-representations. The details are in Sections 2 and 3.2 of our paper. And we add more explanation about it in the updated paper.
> >
> > The "pre-train" is a very common paradigm in current deep-learning field. If lack related foundation, it's difficult to understand our paper. The target readers of our paper are professional researchers, so it's not necessary to detailed explain the "pre-train" paradigm in our paper.
> >
> > **Q4: Why the assumption that daily seasonal effects are dominant? One can construct multiple counter examples where daily effects do not apply (e.g. intraday traffic loads would depend on time of day). Does the model perform ONLY when seasonal effects dominate and are not applicable to general time series datasets?**
> >
> > We did not assume that the daily seasonal effects are dominant. Could you tell which sentence induce you to think so? We'll correct it in the updated paper.
> >
> > We are just recommending the "day" as the patch size, because it's moderate, close to people's habits, and convenient for modeling. We can also choose other patch sizes and you can refer to Q1 of Reviewer Q1u4 or Appendix G of the updated paper for the details.
> >
> > Our model contributes the best prediction on the ER dataset which lacks seasonality. It's enough to prove that our model is applicable to general time series datasets.
> >
> > **Q5:What datasets are being used in the attention patterns for the ablation analysis of E.2.1? This is not stated at all.**
> >
> > PL dataset. And we stated it in the caption of the figure. The criticism: this is not stated at all, is made out of nothing.
> >
> > **Q6:Do the benchmark models get access to the same date/calendar inputs as Dateformer?**
> >
> > Not all benchmark models use the same time features, because these benchmark models may deteriorate if use the same time features input. We have discussed it in Appendix E.2.2. If you read the paper carefully, you cannot ignore it.
> >
> > **Criticism2 : Lack of hyperparameter optimisation – all benchmarks and the dateformer are calibrated using default settings, which makes it difficult to determine whether underperformance is due to improper hyperparams. This is particularly the case for time series data, where optimal hyperparams can vary wildly between datasets?**
> >
> > The criticism is absolutely misunderstanding. We conducted hyperparameter optimization by grid-search. It can sure that all models are full power run. These Transformer-based forecasting models' best hyperparameter setups are generally same as their recommended (not default), except the input lengths of Informer, we adjusted them to 96 and obtained better results. Dateformer is our model, why we use improper hyperparameter to encumber its performance? All these hyperparameter setups are proven by previous works in the long-term series forecasting field, like [1][6][7][8][9].
> >
> > We think conducting hyperparameter optimization is the most basic common sense and not need to be emphasized in a paper submitted to top-level AI conferences like ICLR. But considering your comment, we add the related statement to the updated paper.
> >
> > # Reference
> >
> > [7] Informer: Beyond efficient transformer for long sequence time-series forecasting. AAAI 2021
> >
> > [8] Autoformer: Decomposition transformers with auto-correlation for long-term series forecasting. NIPS2021
> >
> > [9] Pyraformer: Low-complexity pyramidal attention for long-range time series modeling and forecasting. ICLR 2022

---

> > > ### Author Response · Authors · 2022-11-14
> > > **Response to Review Xz1K (Part 3/3)**
> > >
> > > **Criticism3: The two-step training process for Dateformer greatly increases the training time versus other benchmarks – can this be combined to be made end-to-end?**
> > >
> > > Firstly, our model can be made end-to-end. We stated it in the end of Section 3.1 and discussed it in Appendix F. `for some datasets, end-to-end training Dateformer is also appropriate, we discuss it in Appendix F.` If you read the paper carefully, you cannot ignore it.
> > >
> > > Secondly, " two-step training process for Dateformer greatly increases the training time versus other benchmarks" is a false proposition based entirely on conjecture. We use two-step training process to train Dateformer and compare the training time with other baseline models. The results are shown below.
> > >
> > > *Multivariate forecast comparison about training time (metric unit: second).*
> > > |  Datasets   | Ours  | FEDformer | Autoformer | Informer |
> > > |  :----:  | :-----:  | :-----: | :-----: | :-----: |
> > > | ETTh2 | 191 | 1748 | 170 | **84** |
> > > | PL  | **687** | 11508 | 3422 | 2395 |
> > > | Weather  | **137** | 1791 | 502 | 340 |
> > > | Traffic | 328 | 1985 | 419 | **307** |
> > >
> > > It can be seen, although using two-steps training, our model requires less training time than (at least not significantly more than) other baseline models. And facing predictive demands for any number of days, our model only need training once. But other models need to be re-trained separately for each predictive length. This means that our model requires the least total training time. We stated this in the end of Section 3.
> > >
> > > Thirdly, the time series forecasting community doesn't care about training time at all. Because time series datasets are usually small. There will be no intolerable training time gap between models.
> > >
> > > This criticism is untenable and totally unnecessary.
> > >
> > > Thanks for your review. Hope our reply can solve your concerns. If our reply can eliminate the misunderstanding, please raise the score for us. If not, welcome your extra questions and we will answer them in detail. And we will update the paper to be clearer soon, it may make you feel easier to understand.

---

> ### Author Response · Authors · 2022-11-17
> **We are looking forward to your reply**
>
> Dear reviewer,
>
> Did our reply solve your questions? If so, could you please raise the score for our paper? If not, please feel free to let's know your concerns. We would be happy to discuss further with you.
>
> Thanks,
>
> All Authors

---

> ### Author Response · Authors · 2022-11-21
> **We are looking forward to your reply**
>
> Dear reviewer,
>
> Did our reply solve your questions? If so, could you please raise the score for our paper? If not, please feel free to let's know your concerns. We would be happy to discuss further with you.
>
> Thanks,
>
> All authors

---

> ### Author Response · Authors · 2022-12-05
> **Looking forward to your feedback**
>
> Dear reviewer Xz1K,
>
> As there is only one week left until the end of the author-reviewer discussion stage, we would like to ask again whether our paper revisions and responses have addressed your concerns and questions adequately. If not, we would be happy to discuss it further.
>
> Thanks,
>
> All authors

---

> ### Author Response · Authors · 2022-12-09
> **Please be responsible and participate in the discussion**
>
> Dear reviewer Xz1K,
>
> As there are only 3 days left until the end of the author-reviewer discussion stage, we would like to ask again whether our paper revisions and responses have adequately addressed your misunderstanding and questions. If not, we would be happy to discuss it further.
>
> We have reminded you 3 times, and we will be very appreciative if you responsibly engage in the discussion.
>
> Thanks,
>
> All authors

---

> ### Comment · Reviewer_Xz1K · 2022-12-10
> **Thank you for your reply**
>
> Thank you for your reply. Following further discussions with other reviewers, I have raised my score slightly. While I do think the idea has promise, I think one question that still remains for me is on what types of global relationships can be extracted purely from date features alone without access to past values of the target -- beyond just its general level.
>
>
> On the hyperparameter tuning comment -- given that ICLR does place a strong emphasis on the reproducibility of experiments, I do think that training methodology should be more than an afterthought, and encourage the authors to fully present their methodology for critique by the wider community -- including what parameters are tuned and their ranges. Furthermore, the decrease in performance for some models when date features are included is slightly inconsistent with the importance placed on them throughout the paper. In implementation details, it states -- "We adopt the hyper-parameters setting that recommended in their repositories but unify the token’s dimension dmodel as 512.", which seems to imply that token dimension was not tuned by grid search as stated below.

---

> ### Author Response · Authors · 2022-12-10
> **Further response to Reviewer Xz1K**
>
> We are very appreciative that you responsibly engage in the discussion and raise the score. Let's further address your extra question and concerns.
>
> **Q1: What types of global relationships can be extracted purely from date features alone without access to past values of the target -- beyond just its general level.**
>
> Firstly, the temporal patterns that are persistent in the training set and related to time features can be extracted purely using date features. For example, in Figure1a we can see every summer, the power consumption of residents will increase sharply due to the temperature rise.
> Additionally, the overall trend of time series influent time series very slowly, we can only observe it from a high global angle. In Figure1a, we can only observe a slightly upward trend in the whole 2 years power load series. If zoom in, it's difficult to observe the overall trend from the local angle.
>
> Most time series that are related to people's activity have the type of temporal patterns, like power load series, traffic flow series, and so on. We can employ time features as a container to distill and store the whole training set's global pattern information.
>
> Secondly, there may be a misunderstanding for you, the global pattern needs to observe past values. But observing the whole training set rather than the lookback window series to distill the global pattern information. The time feature is a container, before the distilling, it's empty so can't represent the time series' global pattern. After the distilling, we believe the container has stored the time series' global information and hence can represent the time series' global pattern.  We have emphasized in Section 3.1
>
> The time feature may be unsuitable for some global patterns that are not related to time(maybe exists).  In further works, we try to look for other types of containers to better distill and store the global pattern information. But to our best knowledge, we are the first time series forecasting work that explicitly utilizes containers to distill the whole training set's global pattern information. We are confident that the paper provided a brand-new long-term series forecasting paradigm. It is harsh to require absolutely perfect all details for such pioneering work.
>
> **Q2: Hyperparameter tuning.**
>
> For these Transformer-based baseline models, we conduct the grid-search around the hyperparameters recommended by their papers. The best hyperparameters setups are generally the same as those recommended in their papers except the input length of Informer. You can rest assured of the hyperparameter settings of these models. **And you can refer to the reported results in their own papers, it's generally the same level as our reported. This is best reflecting evidence. If any author accuses and proves that we deliberately weaken their model to report the results, we must voluntarily withdraw our paper.**
>
> In Pyraformer's code repository, the token dimension $d_{model}$ for Pyraformer is 256 for some predictive cases. And other cases are 512. For all other Transformers, the $d_{model}$ are unified 512 in their code repository. It's almost a convention for the long-term series forecasting community. For fairness, we unify the cases of 256 to 512. Theoretically, this replacement will not cause performance damage, because 512 > 256. And we didn't find any performance damage in the replacement. The dimension of 512 is absolutely enough for Pyraformer to model each time-step value and has been proved by many other Transformers.
>
> About reproducibility, we promise that will release the code soon. You can check our reported results in the released code.
>
> **Q3: Furthermore, the decrease in performance for some models when date features are included is slightly inconsistent with the importance placed on them throughout the paper.**
>
> We think the decrease in performance for these models is blamed for their usage style of the time features. The time features just play an auxiliary role and are simply added to each time series value tokens in these models, which makes various time series patterns tangle together and difficult to learn. Our Dateformer is the first time-modeling Transformer that takes the time features as the main modeling objective, so it can consistently benefit from better time features and encodings. We discussed it in Appendix E.2.2 of the paper.
>
> Thank you again for participating in the discussion. If you have any extra question, feel free to let's know. If not, we welcome you to improve the score further.

---

### Official Review · Reviewer_VVpa · 2022-10-25

**Confidence:** 3
**Correctness:** 3
**Technical Novelty And Significance:** 3
**Empirical Novelty And Significance:** 3
**Recommendation:** 6

**Clarity, Quality, Novelty And Reproducibility:**

The paper proposed an interesting and novel idea of learning a date representation for global patterns in time series. However, the readability of the model part needs to be further improved. Some implementation details are shared. And the authors claim that the code will also be shared in the future.

**Strength And Weaknesses:**

This paper proposed a novel framework that tackles the time series forecasting task by distilling the temporal pattern in the date presentations.

Strengths
1. SOTA methods are compared in the paper.
2. Comprehensive experiments are conducted.

Weaknesses:
1. The presentation of the model part is difficult to follow.

**Summary Of The Paper:**

This paper proposed a novel framework that tackles the time series forecasting task by distilling the temporal pattern in the date presentations.

**Summary Of The Review:**

In this paper, the authors proposed an interesting framework that uses date representation as a container to carry the global pattern contained in the time series. Detailed comments are listed below:
1. The connection between different model components and each component's functionality is not clearly addressed.
2. What data is used to calculate equation 1 and the pretrained loss? Do the authors use time series in the training dataset? If so, is the time series for the target day used? Or time series of all time stamps involved in the date-embedding sequence? Is the purpose of DERT to acquire a date representation by building a connection between the training dataset time series and the static date embedding? Can the authors clarify that?
3. The global prediction component is difficult to follow. First, global prediction is drawn to present a learned global pattern. However, this component feeds on the G copies of one day's presentation. So why these G copies can present the global pattern of the entire training set series? Second, what ground truth is used to train this global prediction component separately? Third, can the authors provide a clear definition of the "global pattern" or some examples?

---

> ### Author Response · Authors · 2022-11-05
> **Response to Reviewer VVpa (Part 1/2)**
>
> Thanks very much for your review. We would like to answer your insightful questions to address the remaining concerns.
>
> **Q1: The connection between different model components and each component's functionality is not clearly addressed.**
>
> DERT's duty is to encode dynamic date representations. The date representations acts as contationers to store time series' global pattern information. We stated it in the begining of Section 2:`To facilitate distilling training set’s global information, we should establish appropriate time representations as the container to store it.`
>
> Then, we use the global prediction component to distill the whole training set's global pattern information to the containers. During inference, we can draw a global prediction from distilled global information. We stated it in the begining of Section 3.1:`We distill whole training set’s global information and store it in date-representations. Then, in the prediction stage, we can draw a global prediction from therein to represent learned global pattern.`
>
> The local prediction component is used to learn time series' local pattern information from input lookback window. We first eliminate the learned global pattern from lookback window time series ground-truth, the $Series\ Residuals$ contains only time series' local pattern information. We stated it in the begining of Section 3.2:`To better capture the local pattern from lookback window, we eliminate the learned global pattern from lookback window series.`
>
> Add the global and local prediction together to produce the final prediction. As stated in equation 5: $Final\ Prediction = Global\ Prediction + Local\ Prediction \ (5)$
>
> Dateformer exploites these components to automatically conduct above procedures and fuses them into the final predictions, just like a scheduler. We stated it in the bottom of page 5:`For multi-day forecast, we need to encode multi-day date-representations and produce their corresponding global predictions and local predictions. Thus, we design Dateformer to automatically conduct these procedures and fuse them into the final predictions, just like a scheduler.`
>
> **Q2: What data is used to calculate equation 1 and the pretrained loss? Do the authors use time series in the training dataset? If so, is the time series for the target day used? Or time series of all time stamps involved in the date-embedding sequence?**
>
> The training set's time series patches that are split by day are used to calculate equation 1 and the pretrained loss. We only pick the target day’s date-representation out, so only target day's time series patch is used to calculate pretrained loss. As stated in the paragraph under Figure 2: `After encoding, we only pick the target day’s date-representation out and attach liner layers to it. These linear layers act as the implementations of 2 pre-training tasks we design.`
>
> **Is the purpose of DERT to acquire a date representation by building a connection between the training dataset time series and the static date embedding? Can the authors clarify that?**
>
> DERT's duty is to encode dynamic date representations. Your understanding is right.

---

> > ### Author Response · Authors · 2022-11-05
> > **Response to Reviewer VVpa (Part 2/2)**
> >
> > **Q3: Why these G copies can present the global pattern of the entire training set series? Second, what ground truth is used to train this global prediction component separately? Third, can the authors provide a clear definition of the "global pattern" or some examples?**
> >
> > Firstly, please note that the G copies are added sequential positional encodings where G denotes time-steps number everyday. This is to refine the date representation to the time representation that in a finer time scale. We train the global prediction component on training set's time series ground-truth to distill training set's global information and store in these time representations.
> >
> > As stated in the Paragraph 6 of INTRODUCTION: ` time is one of the most important properties of time series and plenty of series characteristics are determined by or affected by it.` So, we try to take time as containers to distill and store the entire training set's global information. The separately training of global prediction component represents the distilling procedure. Before separately training the global prediction component, the container is empty(FFN of the global prediction component is not well trained) and can't present global pattern. But after that, we believe the global pattern information is distilled to the container so we can draw a global prediction from therein during inference. We didn't feed rencent lookback window series into the global prediction component, so it can't learn shor-term local pattern, and only global pattern information is distilled. Table 2 and Figure 6 also prove it.
> >
> > The global pattern includes but not limits to overall trend and long-term existing seasonality of time series. Figure1a is a good example: models that only observe the zoom-in window as input will learn a upward trend; But if zoom out, we know that is a yearly seasonsality. Meanwhile, we can find a upward overall trend on the 2 years power load time series. Reviewer Q1u4 also appreciates the example.
> >
> >
> > Thanks very much for your insightful review. Please let us know if you have extra questions.

---

> ### Author Response · Authors · 2022-11-21
> **We are looking forward to your reply**
>
> Dear reviewer,
>
> Did our reply solve your questions? If so, could you please raise the score for our paper? If not, please feel free to let's know your concerns. We would be really happy to discuss further with you.
>
> Thanks,
>
> All authors

---

> ### Author Response · Authors · 2022-12-11
> **Final inquiry**
>
> Dear reviewer VVpa,
>
> As there is only last 1 day left until the end of the author-reviewer discussion stage, we would like to ask again whether our paper revisions and responses have addressed your concerns and questions adequately.  If so, could you please raise the score for our paper? If not, please feel free to let's know your concerns. We would be happy to discuss further with you.
>
> best regard,
>
> All authors

---

### Author Response · Authors · 2022-11-15
**General Update**

Dear all reviewers and ACs,

Thanks very much for your review. We really appreciate your insightful questions and constructive suggestions. In addition to the respective responses, we have prepared a revised paper that attempts to address all the concerns and to adopt these valuable suggestions completely.
The updated paper has been uploaded, **major changes are highlighted in blue.**

Following the suggestions from Reviewer atRd, we add basic comparison about local prediction components with generative or aggregated style in Appendix M. And we add more detailed explanation about Figure1 and how to train global prediction component.

Following the suggestions from Reviewer Q1u4, we supplement comparative experiment about different patch sizes and model's complexity analyses. In addition, Reviewer Q1u4 is very responsible, professional, and respectable.  He/She gave many detailed suggestions to help us improve the paper to be clearer and well-organized. We sincerely thank him/her for the very helpful reviews. If there is Best Reviewer Award in ICLR, we recommend Reviewer Q1u4.

Referring to the reviews from Reviewer Xz1K, we add more explanation about time series' global pattern information to Figure1a and more details about pre-training DERT. In addition, we add statements about how to conduct hyperparameter optimization.

Following the suggestions from Reviewer VVpa, we take Figure1a as an example to add more explanation about time series' global pattern and how to train DERT and global prediction component. We re-write the model part to make it clear, easier to follow and emphasize how to distill time series' global information.

Please let us know if you have extra questions.

Thanks,

All authors

---

### Author Response · Authors · 2022-12-11
**Summary of Rebuttal**

Dear PCs, ACs, and SACs,

Thank you very much for your efforts in this review, on the whole, it was a pleasant experience. Reviewers expressed their recognition of our work and gave much valuable advice to help us improve the paper.

Reviewer atRd recognized our research direction and gave many professional suggestions to improve the paper to be more solid and clear. The reviewer also inspired our other research directions in future works.

Reviewer Q1u4 appreciates our work and recommends this paper be accepted. The honorable reviewer gave us a lot of useful, detailed, and professional suggestions to help us improve the paper a lot. We sincerely thank the reviewer and commend the reviewer to ICLR.

Reviewer Xz1K has some misunderstanding about our paper. After discussion, we clarified the misunderstanding, so the reviewer raised the score.

Reviewer VVpa appreciated the SOTA performance of our work and the comprehensive experiments. The reviewer also suggested that we should improve the writing of the model part, and we followed the suggestion to rewrite the model part to be clearer.

Overall, the biggest concern of all reviewers for our paper is that the writing is not clear enough. And we have improved this and have been recognized by the reviewers.

***

**However, we think reviewers underestimated the novelty of our work. In addition to what the reviewers got, we would like to emphasize the following 3 points.**

Firstly, beyond a new model, the paper proposed a brand-new time series forecasting paradigm: employ some types of containers(such as time features in the paper) to distill and store time series datasets' global information, thereby grasping time series' long-term global pattern, at a small cost. In future work, we can look for other types of containers or other architectures of the local prediction component, and we are going on.

Meanwhile, we also explore a different style to leverage time features in time series forecasting. Before the paper, time features always play a minor role in time series forecasting works, sometimes even completely ignored, so as to previous various time series forecasting Transformers. We take the time feature as the main modeling objective and achieved good results, which reminded the time series community to explore appropriate styles to leverage time features of time series. We mentioned the idea repeatedly in the paper and conducted experiments to prove that the previous style is somewhat inappropriate, details in Appendix E.2.

Finally, since Informer was published, dazzled modifying Transformer is now popular in the long-term series forecasting community. Our Dateformer proved the vanilla Transformer also can perform well, which may induce the community to explore different routines utilizing Transformer. We note that we are not the only ones to rethink Transformer in time series forecasting. Similar attempts have been made in some concurrent works.

***
This contribution experience is pleasant. We sincerely thank all the participants who helped us improve this work.
Once again, we pay tribute to the reviewers, ACs, SACs, PCs, and all participants.

Best regard,

All authors

---

### Decision · Program_Chairs · 2023-01-20

**Decision:**

Reject

**Justification For Why Not Higher Score:**

see above

**Justification For Why Not Lower Score:**

see above

**Metareview: Summary, Strengths And Weaknesses:**

There was an extensive discussion with the reviewers about this paper but all reviewers converged to the point where they were convinced that the paper was not ready for publication in its current form. The core issues were summarized by the reviewers their recent updates, but we give a very brief recap of the issues here.

1) Unclear presentation and writing. This should have been addressed in the original submission. Note that when reviewers find a paper unclear, the best thing the author should do in the next submission is focus on improving this aspect of the paper.

2) Need to be much clearer, detailed, and revealing specifically about what hyperparameters were tuned, their ranges, and how much effort went into this, and other specifics about how extensive hyperparameters were tuned relative to baselines.

3) The underlying statistical assumptions of the Dateformer should be discussed and analyzed. What specific statistical model of the data would be true for it to be possible to separate date from other content in a way that leads you to get improved results. This is an issue where it is not always enough to show improved results, to contribute to the scientific machine learning community we need to have a deep understanding of why such improvements occur very precisely, and ideally mathematically, and when we might expect such improvements to occur in the future, and under what scenarios might this approach not work so well. For example, details regarding the decrease in performance for some models should be very clearly spelled out in the original submission.


Overall, we think that there is merit in the approach, and if you work carefully on the above, you will have an accepted paper at the next conference.



**Summary Of Ac-Reviewer Meeting:**

see above